# Generating high quality libraries for DIA MS with empirically corrected peptide predictions

Brian C. Searle [1,2✉], Kristian E. Swearingen [1], Christopher A. Barnes[3], Tobias Schmidt [4], Siegfried Gessulat [4,5], Bernhard Küster [4,6] & Mathias Wilhelm [4]

Data-independent acquisition approaches typically rely on experiment-specific spectrum libraries, requiring offline fractionation and tens to hundreds of injections. We demonstrate a library generation workflow that leverages fragmentation and retention time prediction to build libraries containing every peptide in a proteome, and then refines those libraries with empirical data. Our method specifically enables rapid, experiment-specific library generation for non-model organisms, which we demonstrate using the malaria parasite *Plasmodium falciparum*, and non-canonical databases, which we show by detecting missense variants in HeLa.

[1] Institute for Systems Biology, Seattle, WA, USA. [2] Proteome Software, Inc., Portland, OR, USA. [3] Novo Nordisk Research Center Seattle, Inc., Seattle, WA, USA. [4] Technical University of Munich, Freising, Germany. [5] SAP SE, Potsdam, Germany. [6] Bavarian Center for Biomolecular Mass Spectrometry, Freising, Germany. ✉email: bsearle@systemsbiology.org

Data-independent acquisition (DIA) mass spectrometry (MS) is a powerful label-free technique for deep, proteome-wide profiling[1,2]. With DIA, mass spectrometers are tuned to systematically acquire tandem mass spectra at regular retention time and $m/z$ intervals, freeing the method of the intensity-triggering biases introduced by data-dependent acquisition (DDA). To accomplish this, precursor isolation windows are widened such that multiple peptides are usually co-fragmented in the same MS/MS scan. DIA methods generally identify peptides with library search engines[3–5] using experiment-specific spectrum libraries[6] from DDA experiments. In peptide-centric searching[7], library entries are scored according to retention time, such that the best-scoring time point for each peptide is reported. Only peptides present in the libraries can be detected, and the peptide detection reports must be corrected to limit the number of potential false discoveries[8]. Most importantly, these libraries are built at the expense of time, sample, and considerable effort with offline fractionation, especially considering that they are typically not reusable across laboratories or instrument platforms[9].

When experiment-specific spectrum library generation is either impossible or impractical, as is frequently the case with non-model organisms, sequence variants, splice isoforms, or scarce sample quantities, software tools such as Pecan[10] and DIA-Umpire[11] can detect peptides from DIA experiments without a spectrum library by directly searching every peptide in FASTA databases. Gas-phase fractionation[12] (GPF) improves detection rates with these tools[10] by injecting the same sample multiple times with tiled precursor isolation windows, allowing each injection to have narrower windows (and thus fewer co-fragmented peptides) with the same instrument duty cycle. While offline fractionation requires an additional liquid chromatography (LC) step using orthogonal separation modes to online LC-MS, GPF occurs completely within the mass spectrometer, making it both more reproducible and easier to perform. While this method is often prohibitively expensive because it requires enough instrument time and protein content for multiple injections for each sample, the use of multiple GPF injections can be applied just to pooled samples to generate DIA-only chromatogram libraries that make it easier to detect peptides in single-injection DIA experiments[13]. However, even when using GPF, these tools still generally detect fewer peptides than library search engines, which can leverage previously acquired instrument-specific fragmentation and measured retention times.

Recently it has become possible to accurately predict spectra from peptide sequences[14,15], but direct searching of single-injection DIA data has remained problematic, in part due to the false discovery rate (FDR) correction required when considering all possible tryptic peptides in a FASTA database. Proteins show 3–4 orders of magnitude difference in intensity between the best- and worst-responding tryptic peptides[16], and only considering the best-responding peptides in libraries can improve detection rates by lessening the required FDR correction. This approach has been applied by generating independent assay libraries for each DIA injection, either by searching the DIA data directly[5,11], or using paired DIA/DDA experiments[6,17].

Here we demonstrate an approach to generate DIA-only chromatogram libraries from GPF-DIA injections using peptide fragmentation and retention time predictions. This method creates empirically corrected libraries that sidestep the issues of directly searching predicted libraries, because the GPF-DIA injections use the same acquisition parameters, chromatographic conditions, and sample matrix as quantitative single-injection DIA experiments. We observe improved peptide detection rates when searching these empirically corrected libraries over searching sample-specific DDA libraries.

Empirically corrected libraries are built directly from protein sequence databases, allowing our workflow to enable experiments that identify protein-level genetic variants and quantify peptides from non-model organisms.

## Results

**Empirically corrected libraries from peptide predictions.** Here we report on a DIA-only workflow that produces higher-quality libraries than those generated by DDA while simultaneously supplanting the need for any offline fractionation. Our workflow (Fig. 1, Methods) uses a recently developed deep neural network, Prosit[14], to generate a predicted spectrum library of fragmentation patterns and retention times for every +2H and +3H tryptic peptide in a FASTA database, with up to one missed cleavage. Fragmentation prediction in Prosit adjusts based on normalized collision energy (NCE), and we tune the NCE parameter for each peptide charge state to account for DIA-specific fragmentation.

Building on the chromatogram library method[13], we make a pool of sub-aliquots from a representative subset of biological samples in our experiment. In addition to analyzing each biological sample using single-injection DIA (typically 4- to 12 $m/z$-wide precursor isolation windows after staggered-window demultiplexing[18], depending on the instrumentation) with a 90-min gradient, we collect six additional GPF-DIA acquisitions (typically 2 $m/z$-wide precursor isolation windows after demultiplexing, regardless of instrumentation) of the sample pool using the same gradient. Considering column washes, these GPF-DIA acquisitions take approximately 12 total additional hours of MS acquisition. We find that single-injection DIA can benefit from tuning the precursor isolation window to suit the mass spectrometer acquisition rate. However, GPF-DIA acquisitions tend to be ion population-limited, and narrowing precursor isolation windows below 2 $m/z$ (after demultiplexing) does not improve detection rates (data not shown). In part, this may be related to smaller windows breaking up isotopic envelopes, resulting in lower overall sensitivity.

We search the GPF-DIA acquisitions of the pool against the predicted spectrum library using EncyclopeDIA. Searching with the predicted spectrum library has multiple disadvantages over experiment-specific libraries. First, correcting for false discoveries requires more stringent thresholds when considering every possible peptide in a proteome, rather than just those previously detected in a pooled sample. Secondly, while Prosit typically produces higher-quality spectrum libraries with deep learning than other, more conventional approaches[19], the predictions are less accurate than experiment-specific libraries generated on the same instrumentation. However, these two disadvantages are mitigated by the use of GPF-DIA with precursor isolation windows as narrow as those used in targeted parallel reaction monitoring[20] (PRM) or DDA.

Finally, we use the detections made with GPF-DIA to construct a new, empirically corrected library removing the disadvantages of the predicted spectrum library. Assuming that virtually every consistently quantifiable peptide in an experiment is detectable from the pool using GPF-DIA, we filter the predicted library to remove peptides that cannot be found in the pool. In addition, we select the highest-scoring (and therefore easiest to detect) charge state for each peptide and remove other, lower-scoring charge states from the library. Then, for each identified peptide, we calculate the aggregate peak shape across all of the identified fragment ions and extract fragment peak area intensities for all possible B- or Y-type ions that correlate with this shape. Since the GPF-DIA injections are performed using the same instrumentation setup as the single-injection DIA injections, we use these intensities as the fragmentation patterns in the empirically

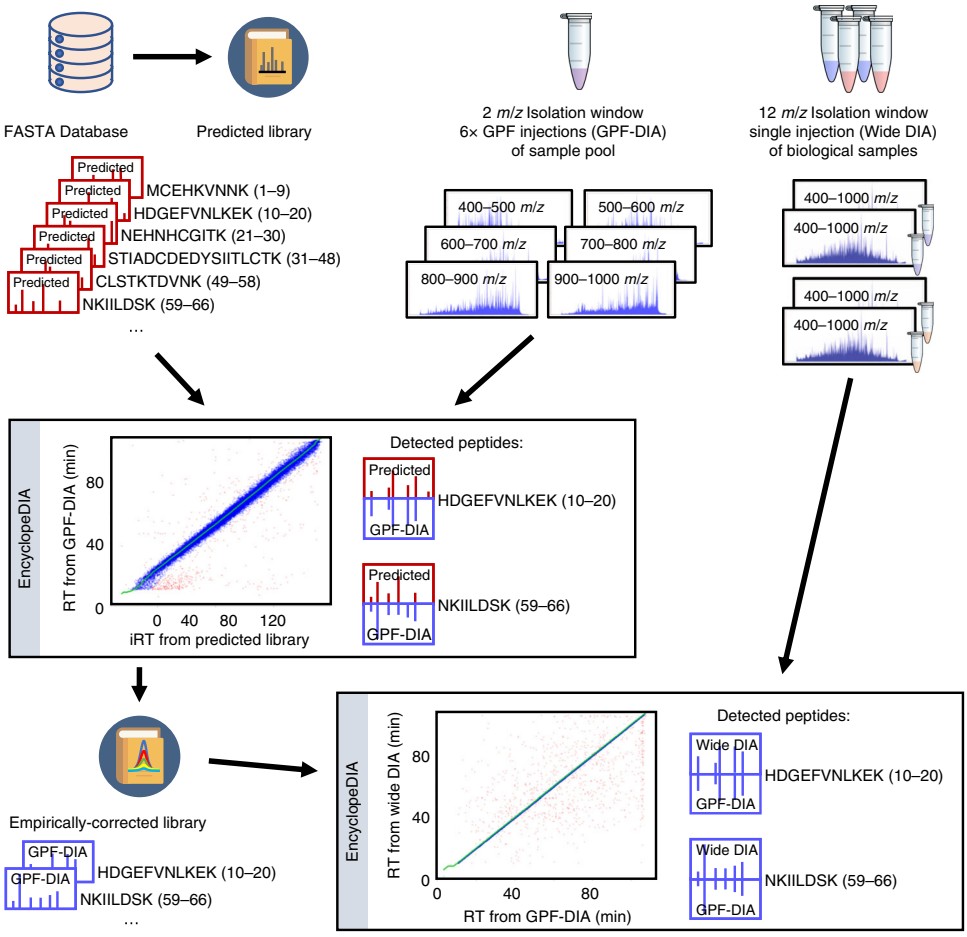

**Fig. 1 Workflow for generating empirically corrected libraries.** Fragmentation patterns and indexed retention times (iRTs) are generated with Prosit for all possible tryptic peptides in a FASTA database, and these predictions are compiled into a predicted spectrum library. In this example, peptides from CDPK2 are shown with start/stop indices within the protein indicated in parentheses (red predicted spectra). We use EncyclopeDIA to search GPF-DIA acquisitions of a sample pool with that library, and peptides detection results are compiled into a experiment-specific, empirically corrected library. This new library contains fragmentation patterns and retention times extracted from the GPF-DIA data for only the detected peptides (blue empirical spectra). Since GPF-DIA and single-injection DIA have the same instrumentation and on-column matrix, retention times and fragmentation patterns in the empirically corrected library are more closely aligned than the original predictions.

corrected library. Similarly, we use the time point of the apex intensity of the aggregate peak shape as the retention time in the new library. We find that while peptide ordering on the same HPLC platform with the same column and method is typically very high, we still benefit from retention time alignment to account for fluctuations in run-to-run chromatography stability. In addition, we perform the six GPF-DIA injections of the pool near the middle of an experiment after at least one full set of biological replicates, to limit variability caused by column (re) conditioning to a new proteome composition. We recommend reacquiring the six GPF-DIA injections if the column or gradient change while an experiment is being conducted.

**Validating the empirically corrected library methodology.** We first applied our method to analyze a yeast tryptic digest on a Thermo Fusion Lumos MS. After column conditioning, we acquired four replicate injections of single injection 400–1000$m/z$ DIA using 4$m/z$-wide windows (after demultiplexing) at 20 ms ion injection time. We followed this by six GPF-DIA injections from 400 to 500$m/z$, 500 to 600$m/z$, 600 to 700$m/z$, 700 to 800$m/z$, 800 to 900$m/z$, and 900 to 1000$m/z$ with 2$m/z$-wide windows (after demultiplexing) at 60 ms ion injection time. Using

a Uniprot *Saccharomyces cerevisiae* protein database (6729 entries), we produced a predicted spectrum library containing 456,511 total entries with 320,150 unique peptide sequences, assuming an NCE of 33. After empirical correction, the new library contained 64,597 unique peptide sequences from 4464 protein groups at a 1% peptide and protein FDR.

Although searching single-injection DIA acquisitions directly with predicted spectrum libraries is highly dependent on prediction accuracy, we found that our workflow produced high-quality libraries even if the predictions were not precisely tuned for the instrument, which suggests broad cross-platform applicability. We demonstrated this by modulating the NCE setting in Prosit (Fig. 2a) and comparing with two libraries: (1) an experiment-specific offline high-pH reversed-phase (HpH-RP) fractionated DDA spectral library containing 39,612 unique peptide sequences acquired at the same time as the DIA experiment and (2) a sample-specific offline SCX fractionated DDA spectral library containing 45,987 unique peptide sequences from another study. Even across a wide range of NCE settings, searching GPF-DIA spectra produced between 33% and 60% larger empirically corrected libraries than we could obtain from the experiment-specific 10-fraction HpH-RP fractionated DDA library (Fig. 2b). Interestingly, the optimal NCE setting for Prosit

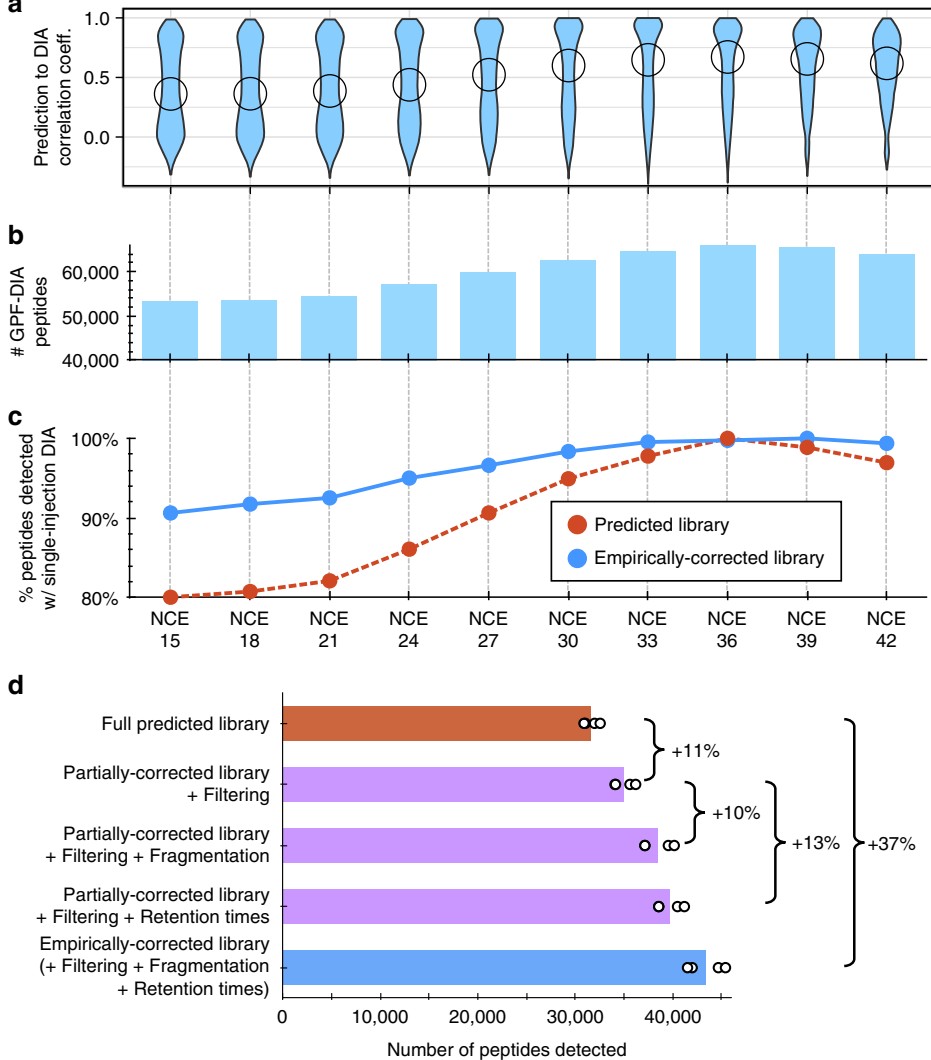

**Fig. 2 Evaluating empirically corrected libraries made with peptide predictions. a** Violin plots showing spectral correlation between predicted library for yeast peptides and single-injection DIA ($N = 1$) at various NCE settings (circles indicate medians) when the instrument was configured for NCE = 33. **b** The total numbers of empirically corrected library entries detected from GPF-DIA at various NCE settings ($N = 1$). **c** The fraction of peptides detected in single-injection DIA ($N = 1$) relative to the optimal NCE for empirically corrected libraries (blue line) are more consistent than predicted libraries (red dashed line) across a wide range of NCE settings. **d** The number of yeast peptides detected at 1% peptide FDR in single-injection DIA acquisitions ($N = 4$) using the NCE = 33 chromatogram library, where either the retention times or fragmentation patterns have been switched with the predicted Prosit values (purple bars). Compared to the predicted spectrum library search (red bar), an 11% increase comes from simply using a narrowed peptide selection. DIA-based retention times and fragmentation patterns provide a 13% and 10% increase over this, respectively. Comparing the empirically corrected library and the predicted library detections (blue bar, 37% increase), these percentage gains appear to be nearly multiplicative (i.e., 111% × 110% × 113% = 138%), indicating that all three factors are independent and of roughly equal importance. Source data are provided as a Source Data file.

was 36 (not the instrument method-specified 33), which likely reflects calibration drift and variation across Orbitrap instruments[21]. Fewer peptides will be detectable in both single-injection DIA and GPF-DIA data at incorrect NCE settings. However, since there is less interference in GPF-DIA, these detection rates do not drop as quickly. After empirical correction, the library will contain fragmentation patterns observed in the GPF-DIA data rather than the original library tuning parameters (Supplementary Fig. 1), and any peptide that can be detected in the GPF-DIA data will be easier to detect in single-injection DIA. In this way, the GPF-DIA functions as a calibration step that corrects the Prosit NCE setting, making searches of single-injection DIA less sensitive to NCE drift or other sources of prediction inaccuracies after empirical correction (Fig. 2c).

Retention time is affected by chromatographic conditions, but also by matrix effects. Single-injection DIA and DDA both measure peptides within the full matrix. Offline fractionation, such as SCX or high-pH reversed-phase, change the matrix by fractionating the peptide mixture into multiple samples. Consequently, each peptide sees a different matrix as it elutes relative to the single-injection injections, causing errors in retention time estimates. Unlike offline fractionation, GPF-DIA uses the quadrupole for fractionation, maintaining the same full matrix complexity of single-injection DIA and improving retention time consistency. This process produced libraries with better retention time accuracy (80% of peptides within 35 s) than both the predicted (80% within 5.4 min) and fractionated DDA libraries (80% within 4.6 min), even when the DDA libraries were acquired

on the same instrument (Supplementary Fig. 2). Coupled with smaller library size and improved fragmentation patterns (Supplementary Fig. 3), these three factors had roughly equal and orthogonal improvements over directly searching single-injection DIA with predicted libraries (Fig. 2d).

We were interested to determine if empirically corrected libraries could be reused for different experiments. To test this, we reanalyzed yeast datasets[13] from a Thermo QE-HF MS at a different location using the empirically corrected library generated in this study on a Thermo Fusion Lumos MS. We found we were able to detect more peptides using the empirically corrected library than could be detected by analyzing the same data with a Prosit-predicted library or FASTA-only approach using Pecan. However, even better results were produced if the data were

analyzed with a library built from GPF-DIA injections collected on the same instrument (Supplementary Fig. 5). In this case, collecting additional GPF-DIA injections and building an empirically corrected library for each experiment improved peptide detection rates by 30%.

While HpH-RP and SCX fractionated DDA produced similar-sized libraries, these libraries draw from different pools of peptides (Supplementary Fig. 4), demonstrating that combining both fractionation methods is necessary for building comprehensive DDA libraries. We observed that searching a combined HpH-RP and SCX library produced more detections in single-injection DIA datasets than either DDA library individually, but that overall, searching single-injection DIA acquisitions with an empirically corrected library detected 31% more yeast peptides (Fig. 3a). Both

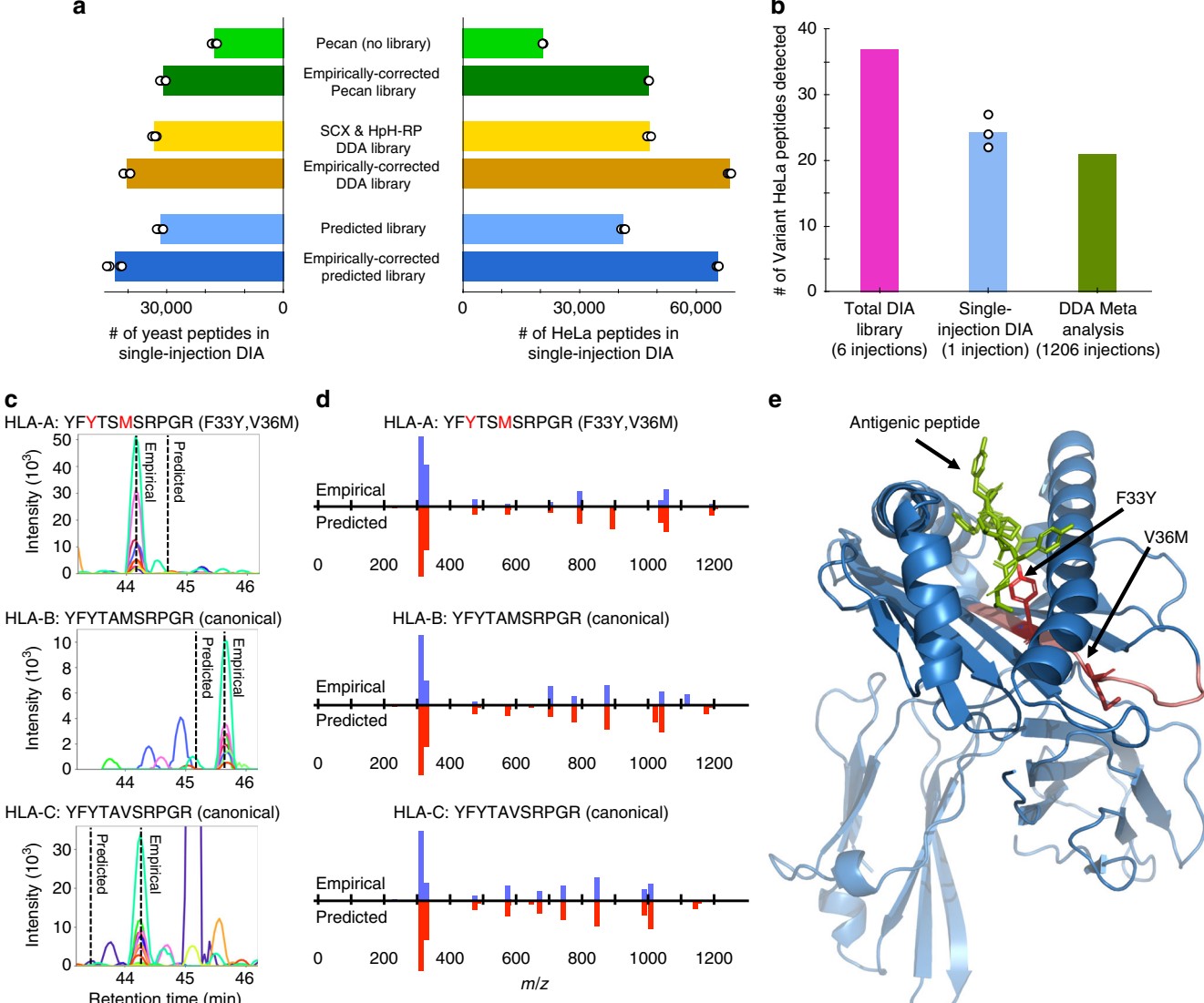

**Fig. 3 Empirically corrected libraries improve peptide and missense variant detection rates. a** The average number of yeast ($N = 4$) and HeLa ($N = 3$) peptides in single-injection DIA detected using library-free searching with Pecan, searching with a combined SCX fractionated and high-pH reverse-phase fractionated DDA spectrum library, or searching with a predicted spectrum library, before or after empirical correction with the chromatogram library method. Match-between-runs was not enabled for any search strategy so that replicates are independent measurements demonstrating the technical variability for each approach. **b** The number of HeLa missense variants detected in the total DIA library, single-injection DIA ($N = 3$), or a meta-analysis of 40 published DDA experiments[27], each filtered to a 1% peptide/protein FDR. **c** Retention time predictions differ somewhat from empirical data from GPF-DIA for homologous peptides: YFYTSMSRPGR from HLA-A (F33Y,V36M), YFYTAMSRPGR from HLA-B, and YFYTAVSRPGR from HLA-C. **d** Relative +1H y-type and b-type fragmentation patterns above 200 $m/z$ for the same peptides are shown as butterfly plots with empirical intensities (blue) above predicted intensities (red). **e** All three peptides are in the HLA peptide binding/presentation region, as indicated by YFYTSMSRPGR (red) in HLA-A (blue) relative to an antigenic peptide (green) in PDB structure 1AO7. Source data are provided as a Source Data file.

Pecan and the combined HpH-RP and SCX library can also be calibrated using GPF-DIA with the chromatogram library method. Searching single-injection DIA datasets with the empirically corrected predicted library outperformed searching the Pecan-based chromatogram library, and detected a comparable number of peptides to searching the DDA-based chromatogram library without any additional DDA measurements or offline fractionation.

**Finding missense variants with non-canonical libraries.** Most publicly available spectrum libraries[22–24] are built by searching millions of spectra against a canonical genome. Our approach facilitates the analysis of sequence variants and splice junctions by simplifying exome-specific library construction, and we demonstrate this by analyzing HeLa-specific missense sequence variants. RNA-seq expression across different HeLa strains[25] suggests that 12.4k genes containing 127 missense variants determined by the COSMIC[26] cell line project are typically expressed (≥10 counts, ≥0.1 RPKM). Recently a custom database including COSMIC variants was used to reanalyze 40 public HeLa datasets containing 1206 total DDA acquisitions[27], globally detecting 21 missense variants. Using this custom database, we built an empirically corrected library for a previously published Thermo Q-Exactive HF (QE-HF) HeLa dataset[13], producing 37% more peptides than the 39-injection DDA library originally used in that analysis (Fig. 3a) and mirroring the results from the yeast experiment. This library contained 7484 protein groups including 37 missense variants (Supplementary Data 1, Supplementary Fig. 6), representing approximately 30% of the expected expressed variants reported by COSMIC. With the library we detected an average of 24 missense variants from single-injection DIA acquisitions (Fig. 3b).

While in general the additionally detected variant-containing peptides will not greatly affect the overall number of quantitative measurements, quantifying key peptides in specific genes, such as the peptide antigen binding region of HLA[28] (Fig. 3c–e), can have a profound effect on biological interpretation. All of the variants detected in this analysis are expected from the genomic data presented in COSMIC. While improved fragmentation and retention time accuracy in empirically corrected libraries can buffer search engines from over-reporting missense variants, care must be taken when attempting to distinguish between homologous sequences with DIA. Heterozygous peptides with similar retention times, such as those associated with G623S from the kinase EEF2K (Supplementary Fig. 7a), can share fragment ions (Supplementary Fig. 7b) and must be localized as if they contained post-translational modifications (PTMs)[29,30] if they fall in the same precursor isolation window.

**Detecting and quantifying peptides from non-model organisms.** We further applied our workflow to analyze cultures of human red blood cells (RBCs) infected with *Plasmodium falciparum*, the parasite responsible for 50% of all malaria cases. We performed DDA and DIA on late-stage (stage IV/V) gametocytes, the form of the parasite that is transmitted from an infected human to the mosquito vector that spreads the disease. Using a QE-HF MS, we produced a 3378-protein empirically corrected library, increasing the previously known proteome[31] size by 58% (Fig. 4a) while only missing 2.5% of proteins detected with other methods. To estimate a null level, we performed the same approach of searching for *P. falciparum* peptides on GPF-DIA injections of uninfected RBCs and found that the library produced zero peptide and protein detections.

Using the empirically corrected library, we measured 2740 proteins in single-injection DIA experiments of *P. falciparum* peptides. Because *Plasmodium* is an obligate parasite, even samples produced by in vitro culture are frequently contaminated with the host proteome. To study the robustness of our method we diluted *P. falciparum* peptides into peptides derived from uninfected RBCs to construct matched-matrix calibration curves[32] for every detected peptide, and found we could detect >80% of the parasite peptides in up to 1:99 dilution (Fig. 4b, Supplementary Fig. 8a). We found that with our method, DIA maintained higher quantification precision and depth at each dilution step than comparable DDA experiments (Fig. 4c, Supplementary Fig. 8b) that were analyzed with MaxQuant[33] with match-between-runs enabled. Leveraging the highly accurate retention times in our empirically corrected library, we found that we could in part recover contaminated *P. falciparum* samples (Supplementary Fig. 9), even when MaxQuant/DDA reported the majority of peptides as missing values.

We analyzed 2444 *P. falciparum* protein groups with at least two peptides (Supplementary Data 2) that were detected in every DIA acquisition of three culture replicates with three technical replicates each (nine acquisitions total). After cross-referencing against the PlasmoDB[34] compendium of 20 different studies at various asexual, sexual, and mosquito stages, we observed 20 proteins previously undetected by mass spectrometry in any life stage of *P. falciparum*. We found an additional 396 proteins that had never been observed in late-stage gametocytes, including 55 previously detected only in immature (stage I/II) gametocytes. Experiments such as these have the potential to redefine the protein expression signature for each stage of the *Plasmodium* life cycle, information that may be vital to identifying targets to disrupt parasite maturation. For example, among those proteins previously undetected in gametocytes is the calcium-dependent

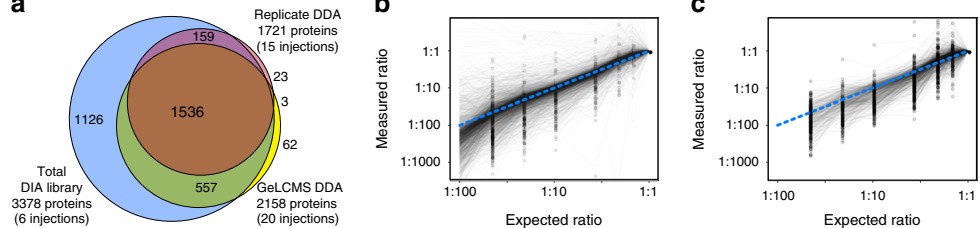

**Fig. 4 Rapid library generation and quantitation of *Plasmodium falciparum* proteins. a** A proportionally sized Venn diagram showing the overlap between proteins detected in the empirically corrected library (blue), the full single-injection dilution curve with DDA (red), and the Stage IV/V mixed male/female sexual *P. falciparum* proteome published by Lasonder et al.[31] from 20× GeLCMS-fractionated DDA acquisitions (yellow). Quantitative ratios of **b** 2740 DIA- and **c** 1491 DDA-detected malaria parasite *P. falciparum* proteins (gray lines) after dilution with uninfected RBC lysate relative to the undiluted sample (N = 1). Dots indicate the lowest dilution level of measurement for each protein, and the blue dashed line indicates the ideal measurement. Empirically corrected libraries enabled DIA quantification of 2152 *P. falciparum* proteins in the last dilution sample (1:99), even when no proteins could be detected or quantified using DDA. Source data are provided as a Source Data file.

protein kinase CDPK2 (PF3D7_0610600). *P. falciparum* parasites lacking CDPK2 develop normally through asexual stages, but male gametocytes are incapable of undergoing exflagellation to become gametes, thereby preventing transmission to the mosquito vector[35]. Our work validates that CDPK2 is indeed present at measurable levels in gametocytes, paving the way to monitor dynamic expression of this kinase over the course of parasite maturation.

## Discussion

In conclusion, empirical correction of predicted spectrum libraries enables rapid experiment-specific library generation for non-canonical proteomes or non-model organisms without off-line fractionation. DDA-based spectral libraries can become stale over time as columns are changed or NCE tuning drifts within an instrument. While the method we propose to create empirically corrected libraries requires an extra 6 GPF-DIA injections for each new experiment, the procedure has the advantage of ensuring that the library is always up-to-date, and even accounts for variation across different instrument platforms.

In addition to DIA applications, this method is applicable for building accurate mass and time tag[36,37] libraries for MS1-only data acquisition strategies, such as BoxCar[38], an approach that forgoes collecting MS/MS and relies on highly accurate mass and retention time indices to identify peptides using match-between-runs. Error rates for match-between-runs peptide detection without MS/MS spectra are often higher than 1% FDR and are hard to estimate without controlled experiments[39]. Errors caused by approaches such as BoxCar are likely exacerbated when the on-column matrix changes, such as between fractionated and unfractionated proteomes. Since our approach builds libraries using the same on-column matrix, retention time tags built with GPF-DIA will likely remove this source of variability.

We also developed a graphical user interface in EncyclopeDIA (Supplementary Note 1) to facilitate making empirically corrected libraries for new proteomes from any FASTA database, which can be converted for external use in both Skyline[40] and OpenSwath[4]. While our interface currently does not support analyzing peptides with PTMs, we believe that when prediction software improves for PTMs, our approach to library building will work for those peptides as well. To encourage the reuse of our method, we have released a growing repository of pre-generated predicted libraries compatible with EncyclopeDIA, Skyline, and Scaffold DIA, which are available for download at ProteomicsDB [https://www.proteomicsdb.org/prosit/libraries].

## Methods

**S. cerevisiae culture and sample preparation.** As described in Searle et al.,[13] *S. cerevisiae* strain BY4741 (Dharmacon) was cultured at 30 °C in YEPD and harvested at the mid-log phase. Cells were pelleted and lysed in a buffer of 8 M urea, 50 mM Tris (pH 8), 75 mM NaCl, 1 mM EDTA (pH 8) followed by seven cycles of 4 min bead beating with glass beads. After a 1 min rest on ice, lysate was collected by piercing the tube and centrifuging for 1 min at $3000 \times g$ and 4 °C into an empty eppendorf. After further centrifugation at $21,000 \times g$ and 4 °C for 15 min, the protein content of the supernatant was removed and estimated using BCA. Proteins were then reduced with 5 mM dithiothreitol for 30 min at 55 °C, alkylated with 10 mM iodoacetamide in the dark for 30 min at room temperature, and diluted to 1.8 M urea, before digestion with sequencing-grade trypsin (Pierce) at a 1:50 enzyme-to-substrate ratio for 16 h at 37 °C. In all, 5 N HCl was added to approximately pH 2 to quench the digestion, and the resulting peptides were desalted with 30 mg MCX cartridges (Waters). Peptides were dried with vacuum centrifugation and brought to 1 µg/3 µl in 0.1% formic acid (buffer A) prior to MS acquisition. All measurements of yeast were performed on the same biological replicate to assess technical variability in the method.

**P. falciparum culture and red blood cell sample preparation.** Human O + erythrocytes (RBCs) were obtained from Valley Biomedical (Winchester, VA; catalog number HP1002O). Three biologically replicate flasks of stage IV/V *P.*

*falciparum* NF54 gametocytes were prepared. Asexual cultures were synchronized with sorbitol and set up at 5% hematocrit and 1% parasitemia. Gametocytogenesis was induced by withholding fresh blood and allowing parasitemia to increase. *N*-acetyl glucosamine was added to media for 4 days beginning 7 days after setup in order to remove asexual parasites. Gametocyte-infected erythrocytes (giRBC) were enriched from uninfected erythrocytes (uiRBC) by magnetic-activated cell sorting at stage III. Stage IV/V gametocytes were collected on day 15 post-setup. Additional uiRBC were also prepared by washing multiple times in RPMI and stored at 50% hematocrit. giRBC and uiRBC cells were lysed in a buffer of 10% sodium dodecyl sulfate (SDS), 100 mM ammonium bicarbonate (ABC), cOmplete EDTA-free Protease Inhibitor Cocktail (Sigma), and Halt Phosphatase Inhibitor Cocktail (Thermo Scientific). Proteins were then reduced with 20–40 mM tris(2-carboxyethyl)phosphine (TCEP) for 10 min at 95 °C and alkylated with 40–80 mM iodoacetamide in the dark for 20 min at room temperature. After centrifugation at $16,000 \times g$ to pellet insoluble material, proteins were purified with methanol: chloroform extraction[41] and dried and resuspended in 8 M urea before the content was estimated using BCA. After dilution to 1.8 M urea, proteins were digested with sequencing-grade trypsin (Promega) at a 1:40 enzyme-to-substrate ratio for 15 h at 37 °C. The resulting peptides were desalted with Sep-Pak cartridges (Waters), dried with vacuum centrifugation, and brought to 1 µg/3 µl in 0.1% formic acid (buffer A) prior to MS acquisition. In addition, several digested peptide mixtures were made by diluting peptides from one flask of giRBC cells with peptides from uiRBC cells at ratios of 1:0, 2:1, 7:8, 4:15, 1:9, 2:41, 2:91, and 1:99 giRBC:uiRBC.

**LC MS (S. cerevisiae).** Tryptic *S. cerevisiae* peptides were separated with a Thermo Easy nLC 1200 on self-packed 30 cm columns packed with 1.8 µm ReproSil-Pur C18 silica beads (Dr. Maisch) inside of a 75 µm inner diameter fused silica capillary (#PF360 Self-Pack PicoFrit, New Objective). The 30 cm column was coiled inside of a Sonation PRSO-V1 column oven set to 35 °C prior to ionization into the MS. The HPLC was performed using 200 nl/min flow with solvent A as 0.1% formic acid in water and solvent B as 0.1% formic acid in 80% acetonitrile. For each injection, 3 µl (approximately 1 µg) was loaded and eluted with a linear gradient from 7% to 38% buffer B over 90 min. Following the linear separation, the system was ramped up to 75% buffer B over 5 min and finally set to 100% buffer B for 15 min, which was followed by re-equilibration to 2% buffer B prior to the subsequent injection. Data were acquired using DIA.

The Thermo Fusion Lumos was set to acquire six GPF-DIA acquisitions of a biological sample pool using 120,000 precursor resolution and 30,000 fragment resolution. The automatic gain control (AGC) target was set to 4e5, the maximum ion inject time (IIT) was set to 60 ms, the NCE was set to 33, and +2H was assumed as the default charge state. The GPF-DIA acquisitions used 4 *m/z* precursor isolation windows in a staggered-window pattern with optimized window placements (i.e., 398.4 to 502.5 *m/z*, 498.5 to 602.5 *m/z*, 598.5 to 702.6 *m/z*, 698.6 to 802.6 *m/z*, 798.6 to 902.7 *m/z*, and 898.7 to 1002.7 *m/z*). Individual samples for proteome profiling acquisitions used single-injection DIA acquisitions (120,000 precursor resolution, 15,000 fragment resolution, AGC target of 4e5, max IIT of 20 ms) using 8 *m/z* precursor isolation windows in a staggered-window pattern with optimized window placements from 396.4 to 1004.7 *m/z*.

For generation of an *S. cerevisiae* spectral library, 80 µg of the same tryptic digests described above were separated into 10 total fractions using the Pierce high-pH reversed-phase peptide fractionation Kit (Thermo, #84868). Briefly, peptides were loaded onto hydrophobic resin spin column and eluted using the following 8 acetonitrile steps: 5%, 7.5%, 10%, 12.5%, 15%, 17.5%, 20.0%, and 50%, keeping both the wash and flow-through. The resulting peptide fractions were injected into the same Thermo Fusion Lumos using the same chromatography setup and column described above, but configured for DDA. After adjusting each fraction to an estimated 0.5–1.0 µg on column, the fractions were measured in a top-20 configuration with 30 s dynamic exclusion. Precursor spectra were collected from 300–1650 *m/z* at 120,000 resolution (AGC target of 4e5, max IIT of 50 ms). MS/MS were collected on +2H to +5H precursors achieving a minimum AGC of 2e3. MS/MS scans were collected at 30,000 resolution (AGC target of 1e5, max IIT of 50 ms) with an isolation width of 1.4 *m/z* with a NCE of 33.

**LC MS (P. falciparum).** Tryptic *P. falciparum* and RBC peptides were separated with a Thermo Easy nLC 1000 and emitted into a Thermo Q-Exactive HF. In-house laser-pulled tip columns were created from 75 µm inner diameter fused silica capillary and packed with 3 µm ReproSil-Pur C18 beads (Dr. Maisch) to 30 cm. Trap columns were created from Kasil fritted 150 µm inner diameter fused silica capillary and packed with the same C18 beads to 2 cm. The HPLC was performed using 250 nl/min flow with solvent A as 0.1% formic acid in water and solvent B as 0.1% formic acid in 80% acetonitrile. For each injection, 3 µl (approximately 1 µg) was loaded and eluted using a 84-min gradient from 6% to 40% buffer B, followed by steep 5-min gradient from 40% to 75% buffer B and finally set to 100% buffer B for 15 min, which was followed by re-equilibration to 0% buffer B prior to the subsequent injection. Data were acquired using either DDA or DIA.

The Thermo Q-Exactive HF was set to acquire DDA in a top-20 configuration with auto dynamic exclusion. Precursor spectra were collected from 400 to 1600 *m/z* at 60,000 resolution (AGC target of 3e6, max IIT of 50 ms). MS/MS were

collected on +2H to +5H precursors achieving a minimum AGC of 1e4. MS/MS scans were collected at 15,000 resolution (AGC target of 1e5, max IIT of 25 ms) with an isolation width of 1.4 $m/z$ with a NCE of 27. Additionally, six GPF-DIA acquisitions were acquired of a biological sample pool (60,000 precursor resolution, 30,000 fragment resolution, AGC target of 1e6, max IIT of 60 ms, NCE of 27, +3H assumed charge state) using 4 $m/z$ precursor isolation windows in a staggered-window pattern with optimized window placements (i.e., 398.4–502.5 $m/z$, 498.5–602.5 $m/z$, 598.5–702.6 $m/z$, 698.6–802.6 $m/z$, 798.6–902.7 $m/z$, and 898.7–1002.7 $m/z$). Individual samples used single-injection DIA acquisitions (60,000 precursor resolution, 30,000 fragment resolution, AGC target of 1e6, max IIT of 60 ms) using 16 $m/z$ precursor isolation windows in a staggered-window pattern with optimized window placements from 392.4 to 1008.7 $m/z$.

**FASTA databases and predicted spectrum libraries.** Species-specific reviewed FASTA databases for *Homo sapiens* (25 April 2019, 20415 entries) and *Saccharomyces cerevisiae* (25 January 2019, 6729 entries) were downloaded from Uniprot. The *Plasmodium falciparum* FASTA database[42] was downloaded from PlasmoDB[34] version 43 (24 April 2019, 5548 entries). The Ensembl-based HeLa-specific FASTA database[27] was downloaded from the ACS Publications website and modified to be compatible with EncyclopeDIA (47,305 entries, including both canonical and variant protein sequences). Each database was digested in silico to create all possible +2H and +3H peptides with precursor $m/z$ within 396.43 and 1002.70, assuming up to one missed cleavage. Peptides were further limited to be between 7 and 30 amino acids to match the restrictions of the Prosit tool[14]. In general, NCE were assumed to be 33 (yeast was processed using NCE from 15 to 42 in 3 NCE increments) but modified to account for charge state. Since DIA assumes all peptides are of a fixed charge, we adjusted the NCE setting as if peptides were fragmented at the wrong charge state using the formula:

$$\text{Adjusted NCE} = \text{NCE} \times \frac{\text{factor(default charge)}}{\text{factor(peptide charge)}}, \quad (1)$$

where the factors were 1.0 for +1H, 0.9 for +2H, 0.85 for +3H, 0.8 for +4H, and 0.75 for +5H and above[43]. After submitting to Prosit, predicted MS/MS and retention times were converted to the EncyclopeDIA DLIB format for further processing. Scripts to produce Prosit input from FASTAs and build EncyclopeDIA-compatible spectrum libraries from Prosit output are available as functions in EncyclopeDIA 1.0.

**DDA data processing.** All Thermo RAW files were converted to .mzML format using the ProteoWizard package[44] (version 3.0.18299) using vendor peak picking. DDA data were searched with Comet[45] (version 2017.01 rev. 1), allowing for fixed cysteine carbamidomethylation, variable peptide n-terminal pyro-glu, and variable protein n-terminal acetylation. Fully tryptic searches were performed with a 50 ppm precursor tolerance and a 0.02 Da fragment tolerance permitting up to two missed cleavages. High-pH reversed-phase fractions were combined and search results were filtered to a 1% peptide-level FDR using PeptideProphet[46] from the Trans-Proteomic Pipeline[47] (TPP version 5.1.0). A yeast-specific Bibliospec[48] DDA spectrum library was created from Thermo Q-Exactive DDA data using Skyline[40,49] (Daily version 19.0.9.149).

*P. falciparum* and RBC DDA data were additionally processed with MaxQuant[33] (version 1.6.5.0) to perform label-free quantitation with precursor ion integration. MaxQuant was configured to use default parameters, briefly fixed cysteine carbamidomethylation, variable methionine oxidation, and variable protein n-terminal acetylation. Fully tryptic searches were performed with a 20 ppm fragment tolerance using both the human and *P. falciparum* FASTA databases, as well as common contaminants and filtered to a 1% peptide-level FDR. Quantification was performed using unique and razor peptides with the match-between-runs setting turned on.

**DIA data processing.** DIA data were overlap demultiplexed[18] with 10 ppm accuracy after peak picking in ProteoWizard (version 3.0.18299). Searches were performed using EncyclopeDIA (version 0.8.3), which was configured to use default settings: 10 ppm precursor, fragment, and library tolerances. EncyclopeDIA was allowed to consider both B and Y ions and trypsin digestion was assumed.

**FDR estimation.** All searches are performed using the target/decoy strategy[50]. As previously described[13], EncyclopeDIA generates decoy peptide sequences by keeping the first and last amino acids in place, but reversing the remaining inbetween sequence. Decoy spectra are generated by moving all fragment ions corresponding to amino acids to the mass appropriate for the new decoy sequence. Each decoy peptide retains the same retention time as the corresponding target peptide. Retention time is only used as a feature (not a filter), so every peptide (decoy or target) can be assigned at any retention time. This approach is designed to give decoys a chance to produce higher scores and better model truly incorrect peptides. EncyclopeDIA search results were filtered to a 1% peptide-level using Percolator 3.1 (refs. [51,52]). Proteins are then parsimoniously allocated to protein groups and filtered to a 1% protein-level FDR.

**Empirically corrected library generation.** Predicted libraries were corrected with EncyclopeDIA using the chromatogram library method described previously[13], and a tutorial for this process is outlined in Supplementary Note 1. Briefly, GPF-DIA injections for a given study were loaded into EncyclopeDIA using the above parameters, where the search library was set to the appropriate predicted spectrum library. Peptides detected by EncyclopeDIA were exported as a chromatogram library.

Percolator is rerun on peptides detected from the GPF-DIA injections to globally filter peptide detections to a 1% FDR. Only peptides detected at a 1% peptide FDR in both the individual GPF-DIA injection and the global analysis are retained for the empirically corrected library. For each detected peptide, fragment ion chromatograms are Savizky-Golay smoothed[53], normalized to the same peak area, and a peptide peak shape is calculated using median smoothing between these chromatograms. A Pearson's correlation score is calculated for every fragment ion indicating the agreement between the overall peptide peak shape and the fragment peak shape.

A peptide entry in a chromatogram library is similar to a peptide entry in a spectrum library in that it contains a precursor mass, retention time, and a fragmentation spectrum. In addition, a chromatogram library entry also contains the peptide peak shape and a correlation score for each fragment ion. This score provides an indication of the likelihood the fragment ion was interfered with in the GPF-DIA injection, with the expectation that it will also likely be interfered with in single-injection DIA as well. This process created a new empirically corrected library containing only peptides found in the GPF-DIA samples, and also retained empirical fragment ion intensities and retention times observed from the DIA data. These libraries were made to be compatible with both EncyclopeDIA and Skyline and were used for downstream analysis of single-injection DIA.

After library generation, FDR estimation for single-injection DIA experiments was performed twice: once at the individual injection level, and again globally across all quantitative samples. For peptide detection experiments, the match-between-runs approach was not used. For the quantitative *P. falciparum* experiments, match-between-runs was applied for peptides not detected in every injection, but only if the peptide was detected at a 1% FDR in the global analysis and at a 1% FDR in at least one individual injection.

Further validation was used for constructing the HeLa-specific library. Here, peptides with similar sequences that fall in the same precursor isolation window can be incorrectly identified by shared fragment ions alone. This class of peptides falls outside of target/decoy-based false discovery estimation and require additional FDR control. Missense variants in the HeLa empirically corrected library were manually validated by checking for variant-specific ions that follow the peak shape. Peptide detections made with no variant-specific ions were considered likely false discoveries and removed from the library.

**Reporting summary.** Further information on research design is available in the Nature Research Reporting Summary linked to this article.

## Data availability

The raw data from the yeast and *P. falciparum* studies are available at MassIVE (MSV000084000 [https://doi.org/10.25345/C5BD2H], ProteomeXchange PXD: PXD017705) and file descriptions are listed in Supplementary Data 3. The raw data from the HeLa reanalysis are available as originally published at MassIVE (MSV000082805 [ftp://massive.ucsd.edu/MSV000082805/]). The source data underlying Figs. 2a–d, 3a, and 4a–c and Supplementary Figs. 2, 3 and 8 are provided as a Source Data file. All other data are available from the corresponding author on reasonable request.

## Code availability

Prosit (https://www.proteomicsdb.org/prosit) and EncyclopeDIA 1.0 (https://bitbucket.org/searleb/encyclopedia) are both available under the Apache 2 open-source license.

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

## Acknowledgements

We thank S. Kappe for insightful discussions, N. Carmago for providing the gametocyte samples, L. Pino for providing yeast lysates, B. Kim for technical assistance, and J. Haskin for editorial assistance. We also thank M. MacCoss, K. Grove, and M. Guldbrandt for providing instrument time. B.C.S. is supported by the Translational Research Fellows Program (TRFP) from the Institute for Systems Biology. K.E.S. is supported by K25AI119229. This work was supported by NIH grant R01GM133981, the German Federal Ministry of Education and Research (BMBF, grant no. 031L0008A and no. 031L0168) and the EU Horizon 2020 grant EPIC-XS (grant no. 823839).

## Author contributions

B.C.S. conceived the study. B.C.S., K.E.S., and C.A.B. performed the experiments. B.C.S., T.S., and S.G. developed the software. B.C.S., B.K., and M.W. supervised the work. All authors wrote and approved the manuscript.

## Competing interests

The authors declare the following competing interests: B.C.S. is a founder and shareholder in Proteome Software, which operates in the field of proteomics. M.W. and B.K. are founders and shareholders of OmicScouts GmbH and msAId GmbH. T.S. and S.G. are founders and shareholders of msAId GmbH. OmicScouts and msAId operate in the field of proteomics. The other authors declare no competing interests.
