## [Peer Review File · Nature Communications]

Reviewers' comments:

Reviewer #1 (Remarks to the Author):

The manuscript "Generating high-quality libraries for DIA-MS with empirically-corrected peptide predictions." by Searle et al. is a very innovative study in which the authors take advantage of sophisticated predicted peptide spectral libraries that can be even further refined by empirical data from DIA chromatogram libraries. This work combines various new modern approaches in working with DIA data so it certainly is a very valuable manuscript. In fact, the discussions about library approaches and DIA data are of great need in the field of DIA proteomics, so this study is very timely. Major points that the authors make are that this approach described here, reduces or eliminates the need for offline fractionation, and as the authors mention from their experiences is more efficient than other approaches, or approaches that search DIA data directly without library (e.g. Pecan). The authors also mention these new high-quality data bases require potentially less stringent FDR or have less false positives.

Comments:

- A minor point: When referring to MS acquisitions I would recommend to change run to acquisition which is a somewhat more formal, more appropriate way to describe this for a written manuscript.
- In the supplement there is a nice description in tutorial style how to use Prosit and how to generate the libraries and process further for the empirical corrections. However it would be nice if in the main manuscript it could be discussed somewhat more in detail how exactly the empirical correction is performed conceptually. This is described very sparsely in the main manuscript while it is the heart to the study. For example, on page 3 the authors mention some of the workflow. The authors mention: "We also replace the retention times and fragmentation patterns in the predicted library with sample- and instrument-specific values found in the 6 runs." Is this a full replacement or an adaptation. Also, if the RT is replaced does that mean that in the lab that will be using the predicted libraries, the user needs to acquire their GPF acquisitions (or chromatographic libraries for empirical correction) right before their real sample DIA acquisitions? Likely not; but what should be paid attention to. Could the authors elaborate in more detail the chromatographic retention time (RT) aspect of the empirical correction. The MSMS fragmentation adjustments seem more intuitive and clear; however as the RT aspects of the correction seem to be so crucial – does it matter at what time the user lab acquires their chromatographic libraries – or will this also work if they acquired those much earlier in time, and then align for that later---what are aspects the user has to pay attention to. Some form of deeper than provided visualization of the workflow would be beneficial.
- In the methods section on page 18, there is a section called "Empirically-corrected library generation" which is very short; again while this is really what the entire paper is based on – it would be good to elaborate a bit more.

- The manuscript then describes several examples where this approach was applied, however, I think for the method of this approach maybe a workflow that is more detailed than the current small Figure 1a would be very helpful. That can bring out key points such as use of the empirical database primarily to detect peak groups; while then the correction uses 'in house' (or also other?) chromatographic libraries so that in the end for the corrected library fragmentation and RT are derived from the chromatographic library solely?
- In Figure 1d the authors state that there is a big discrepancy between peptide detection using either a predicted library or an empirically corrected library particularly at the lower end of the NCE (the authors state that the NCE is not optimized at the lower end), but could this be explained a bit more possibly?
- At the end of the manuscript maybe in the discussion it would be helpful if the authors could indicate what would be the strategies who to implement this workflow into the users own lab. What are the steps the users need to do in order to make this workflow work in their own lab so it becomes a useful tool for users. Would the user need to acquire and generate their own chromatographic library with GPF – would other sample-specific chromatographic libraries generated by the user from their own samples work potentially also (if they are not GPF, maybe a nice variable window – built library would also work) ? Can the users use other people's chromatographic libraries ? What steps can the authors recommend to users how to implement this for their own work – this could be discussed or presented in a schematic overview. Even if not all answers are known, it would be a good thing to discuss.
- As part of this manuscript data is uploaded or being re-used and this is reported; ProSIT and EncyclopeDIA are also mentioned with their weblink – but are the empirically corrected libraries also provided on some repository – this would be good to have.
- In Figure 3 there are some Venn diagrams and overlap of detection numbers using various different approaches; do the authors think this may have to do with use of different instrument platforms (Lumos, QE ? – it gets difficult to keep track what was acquired on what platforms) in some of those used data sets (in the Venn diagram).

Reviewer #2 (Remarks to the Author):

In their manuscript, Searle et al. describe a workflow to generate spectral libraries for data-independent acquisition mass spectrometry-based proteomics (DIA-MS). DIA-MS has been popularized over the past years and to reach optimal performance, it typically requires empirically acquired spectral libraries, frequently spanning dozen to hundreds of additional data-dependent acquisition (DDA) measurements. Previously, strategies have been proposed to alleviate this burden by generating public repositories and this has worked reasonably well for selected model systems. A

bigger issue are applications of DIA-MS to less characterized samples and organisms, where it is typically tedious to acquire such comprehensive spectral libraries.

The authors now combine their previously published spectral library prediction methods with their previously published gas-phase fractionation (GPF) method for library refinement. They demonstrate convincingly that this is a powerful strategy to generate comprehensive spectral libraries that would otherwise be very challenging to acquire. The examples of the HLA variant detection and *Plasmodium falciparum* illustrate clearly the benefits of their workflow for such applications. I believe that their work can have a considerable impact on how spectral libraries are generated for DIA and that it might hopefully replace DDA-based methods in the future. I thus recommend the paper for publication, however, I have a few minor comments and suggestions.

1. Comparison to deep fractionated libraries

Currently, the highest number of detected peptides and inferred proteins with single injection DIA were reported by using heavily fractionated, repository libraries [1]. The authors show the performance of the direct application of a 39-injection DDA HeLa library, which I believe is reasonably comprehensive. In the raw comparison, they also show that this library performs better than the unrefined predicted library. But how does the DDA HeLa library perform when it is refined by the same approach? Does the DDA library contain the majority of peptides/proteins contained within the refined predicted library or do they enable detection of different peptides?

2. Supplementary Figure 3

In their original paper, describing the Prosit method to predict spectral libraries, the authors benchmark the predictions extensively. However, after reading the manuscript here, I was surprised that the authors did not provide an assessment of the decomposed effects of their library refinement workflow in the main text. Supplementary Figure 3 contains all information and I believe it would be useful to convince the readers on the performance of each step if it could be displayed as part of Figure 2.

[1] <https://www.mcponline.org/content/16/12/2296.long>

Reviewer #3 (Remarks to the Author):

The authors propose an approach to generate spectral libraries for analysis of DIA proteomics experiments which proceeds in three main steps: 1) generate a large spectral library of predicted MS/MS spectra using ProSight and a fasta file with protein sequences for the species of interest, 2) acquire new “filtration DIA” data for the sample type of interest and use it to filter the predicted spectral library to contain only matches to the “filtration DIA” runs and 3) use the filtered spectral library to detect peptides in new single-injection DIA runs of the *exact same* sample type. Under these circumstances, the manuscript reports gains in the sensitivity of detection of peptides, proteins and missense sequence variants, but unfortunately it was not possible to thoroughly evaluate these due to the lack of details and supporting information (as described below).

The first major issue with the manuscript is that (1a) the comparisons between DDA and DIA results are nearly impossible to evaluate and (1b) whatever comparisons appeared possible from the deposited data indicate severe biases against DDA that could suffice to explain the gains reported in the manuscript. Regarding (1a), there is no indication of what deposited raw files were used to derive which set of conclusions in the manuscript so there is no way to properly evaluate whether the results are due to the software analysis or instead due to issues with the acquisition protocols. For example, the deposited data does not seem to include any data for the plasmodium dilution experiments and it's completely unclear which MS runs were used to construct the DDA libraries used as point of reference, or which runs were used to derive the sets of search results reported for the DIA/DDA comparisons. See “other comments” at the end of the review for a suggestion of how this information could be properly specified in a revised version of the manuscript. Regarding (1b), even a cursory analysis of the yeast DDA data in your MSV000084000 dataset reported in the paper reveals that the DDA runs have an average of 5 MS2 scans per MS1 scan (much worse than expected for the top-20 configuration described in online methods, average 48k MS2 spectra per file), whereas the single-injection yeast DIA files in the same dataset have an average of 152k MS2 spectra per DIA file, thereby making it much less surprising that DIA could generate more identifications (as it may be simply due to the acquisition settings generating many more MS2 spectra).

Also related to problem (1b) above, the second major issue with the manuscript (2) is that it does not compare to the state of the art DDA acquisition mode – the BoxCar method published by Meier et al in 2018 [PMID:29735998]. The BoxCar approach was clearly shown to substantially increase DDA sensitivity by reducing the dynamic range limitations of MS1 scans, which are critical for the selection of precursors for MS2 acquisition – a problem that seems to significantly constrain the utility of DDA runs used for this manuscript (as discussed for 1b above). The lack of this comparison is even more glaring since the approach proposed in this manuscript uses similar concepts of gas phase separation and staggered window patterns, thereby severely biasing the comparisons of any DDA and DIA results even before consideration of any software or data analysis steps (which are the main point of the submitted manuscript). While the authors could argue that the BoxCar approach is a recent approach that is not yet widely used, that does not suffice to exclude a comparison with this state of the art DDA approach, as otherwise this manuscript would be unfairly comparing state of the art DIA acquisition with what is essentially a decades-old DDA acquisition mode that is already known to be inferior to the DDA state of the art.

The third major issue with the proposed approach is that (3a) it still requires additional MS runs to generate spectral libraries for peptide detection and (3b) it generates spectral libraries that could be extremely sample-specific and thus nearly useless for different sample types, even if from the same species (e.g., different tissues or cell lines). The first part of concern (3a) is the assertion in the manuscript that DDA runs are less useful than DIA runs for pruning predicted spectral libraries. Also related to major issue (2) above, this cannot be properly established without comparing with BoxCar DDA runs to properly assess whether those runs would retain larger subsets of the predicted spectral libraries. In fact, the comparison to DDA should also consider the possibility of retaining predicted spectral library entries if the DDA run (BoxCar or not) contains a LC/MS feature for the precursor m/z within the retention time tolerance, regardless of whether the DDA run generates an MS2 spectrum for the LC/MS feature or not. It would then be important to report how much larger the resulting libraries would be and to assess the impact in loss of sensitivity (if any). The second part of (3a) is that the obvious alternative to requiring additional MS runs (DDA or DIA) would be to use existing spectral libraries as a way to filter precursors in the Prosit predicted spectral library. As such, the authors could filter the yeast predicted spectral library to include only precursors contained in the NIST yeast spectral libraries (>65k distinct precursors over 3 libraries, see <https://chemdata.nist.gov/dokuwiki/doku.php?id=peptidew:cdownload>) or precursors in the PeptideAtlas yeast spectral library (86k distinct precursors, see <http://www.peptideatlas.org/builds/>). Similarly, the predicted spectral library used for the HeLa searches could be filtered using the entries in the PeptideAtlas human spectral library (1,807,958 distinct precursors) or in the MassIVE-KB human spectral library (2,154,269 distinct precursors). In fact, since MassIVE-KB was generated from Thermo/QExactive data and should thus be compatible with the DIA spectra generated for the results comparisons in this study, the manuscript should also report the results of using MassIVE-KB for peptide detection in the DIA runs when adding predicted retention times to MassIVE-KB precursors. In particular, note that this experiment is conceptually equivalent to replacing predicted spectra with MassIVE-KB spectra while retaining the predicted retention times in the Prosit library – this is also important since the authors report that predicted spectra were not optimal and the results improved when replacing them with real spectra (page 4, lines 8-9). As such, it is important to determine whether using real MS2 spectra from other libraries would also perform better than the predicted MS2 spectra.

Regarding issue (3b), the expected utility of the proposed method is directly proportional to its generalization to other sample types so it is important to assess how well the filtered spectral libraries would work for different sample types such as different tissues or different cell lines. A simple way to test this would be to re-search the QExactive data from K562 cell lines used for the original DIA-Umpire paper (file 140120_Lysate_90min_DIA_53.raw), which is readily available in the public dataset PXD001587. This file (and hopefully other DIA runs from other human samples) should be used to show how the performance of the DIA-filtered predicted spectral library compares with that of the spectral libraries filtered to contain precursors from PeptideAtlas or MassIVE-KB (including direct searches with predicted retention times). If it turns out that the public-library approach performs comparably in terms of sensitivity on HeLa runs but is much more generalizable to other types of samples, then the manuscript will need to clearly quantify and justify the expected

gains from requiring additional instrument time to acquire GPF DIA data solely for filtering predicted spectral libraries.

The fourth major issue is that there are many missing details about the detection of HeLa variants and some of the reported results appear to be somewhat nonsensical. First, it is important that post-translational modifications (real or artefactual) be considered as possible alternative explanations for the reported variants. MS2 spectra should be provided in supplementary materials (both reference and variant predicted spectra and real spectra if present in DDA data) and it should be discussed how the distinction from possible modifications is made using only the single-injection DIA data (e.g., how-many/which fragments were used, what mass accuracy was required, show spectra/chromatograms, etc). Second, both reference and variant peptide sequences should be provided in supplementary materials. The submitted variants supplementary table has a number of issues: i) some variants are reported to contain mutations to amino acids that do not occur in the sequence (e.g., "A119T" for peptide #13, but the peptide does not contain Threonine); ii) the alternative interpretation that "Peptide Sequence" would mean "reference sequence" also does not hold since peptide #2 is listed with a mutation "I253T" but contains no Isoleucine; iii) the notation "X-number-Z" for amino acid variants is expected to represent a variation of amino acid X to amino acid Z at coordinate "number" in the protein sequence but the numbers reported in the spreadsheet make no sense in this context. For example, peptide #14 is reported to contain two amino acid substitutions, one at coordinate 934 (C934G) and another at coordinate 1354 (R1354H), which could never occur on the same peptide if the substitutions were over 400 amino acids apart. Since the inconsistency remains even if the coordinates are in base pairs on the original transcript sequence (the substitutions would still be >130 amino acids apart), it is important to correct these to more meaningful amino acid coordinates (preferably on the corresponding UniProt protein sequences). Third, it is surprising for the authors to claim that a single-injection DIA run could detect more variants than 1,206 DDA runs of the same cell line. Since there is a very high chance that this subset of new/surprising variants might be substantially enriched for false discoveries, it is necessary to also list in the supplementary materials table whether each peptide was previously detected or not.

Other comments

- Need to provide supplementary materials with full list of files used for each experiment described in the manuscript. In particular, it is important to describe which files were used for construction of spectral libraries and which files were used to derive each set of DDA or DIA results presented in the manuscript. One simple way to achieve this is to refer to each set of results as experiment X (where X could be a number) and then provide a spreadsheet in supplementary materials linking each raw file to an experiment if the raw file was used to construct the corresponding library (one kind of experiment) or to derive the corresponding DDA or DIA search results (another type of experiment).

- Figure 1d shows that Prosit prediction at NCE 36 was the best match to the experimental NCE 27 used for the MS runs – was that the case for all libraries? If yes then should it be concluded that the Prosit NCE parameters need to be recalibrated?

- The retention time predictions also seemed to be inaccurate (80% of peptides within 5.4 minutes) and the variation between the DDA and DIA retention times also seems very high (80% of peptides within 4.6 minutes), whereas the matches of retention times between GPF runs and single-injection DIA runs was much better (80% of peptides within 35 seconds). What is the rationale for the dramatically better match of retention times between GPF/single-injection DIA runs than between DDA/DIA runs? If all MS runs were generated under the same chromatographic conditions then there should not be such dramatic variation in the retention times between DDA and DIA runs. Of particular importance, how much worse would the search results be if the GPF-DIA-filtered predicted spectral libraries had to deal with the same retention time errors as the libraries derived from DDA data? How can it be quantified how much of the improved performance comes from correcting retention time disparities that should not exist in the first place (i.e., in DDA vs DIA runs; the disparities are understandable in relation to predicted retention times)?

- It was not possible to access the spectral libraries at ProteomicsDB (<https://www.proteomicsdb.org/prosit/libraries>) since the link is not accessible via free anonymous proxy services. All missing data and all full and filtered spectral libraries should be uploaded to a new dataset at one of the ProteomeXchange repositories.

- The proposed approach is conceptually similar to the approach used by MSPLIT-DIA[PMID:26550773] where there is a first stage of analysis that reduces the size of a large spectral library at a fixed FDR and then there is a second stage that uses the reduced library for peptide detection (e.g., see “MSPLIT-DIA library” results reported in Supp. Fig. 9). The approaches should be compared in the manuscript and it should be discussed why the proposed approach is preferable to that of MSPLIT-DIA.

- A more extensive discussion of controls for false discovery rates (FDRs) needs to be included in supplementary materials, especially as it relates to demonstrating that the gains in identifications are not obtained at the cost of elevated FDR. In particular, while the authors do mention that the filtered libraries are generated using EncyclopeDIA at 1% peptide- and protein-level FDRs, it is not clear that any of this holds when running the downstream analyses of single-injection DIA data. This process needs to be described in detail and needs to include a discussion about why and how FDR controls remain valid for single-injection DIA search results. In particular, if the downstream analyses of single-injection DIA data use peptide-centric FDR estimates then it needs to be shown that these are comparable to the spectrum-centric FDR estimates used for the DDA results (otherwise the difference in results could be largely due to just this issue).

- It appears from the manuscript that the Plasmodium single-injection files were searched using only predicted spectral libraries (filtered or not) from Plasmodium sequences. Since the dilution experiments in human red blood cells are obviously also going to contain human peptides, it is necessary to also report results when human sequences are also used a) for the construction of the filtered library and b) for the searches of single-injection dilution runs. In particular, what would be the impact in the loss of sensitivity for plasmodium peptides? Would any DIA fragments/chromatograms switch from plasmodium peptides to blood peptides, especially if searching the same data only against filtered spectral libraries of only blood peptides? These questions are especially relevant in the context of the dependencies that are ignored for peptide-centric analyses that seem to be central to this manuscript so it is necessary to conduct these experiments and to clearly show that the reported gains are not mostly the result of poor FDR controls for false discoveries.

We thank the reviewers for the positive feedback and useful comments on our manuscript, and addressing their comments has improved our manuscript in meaningful ways. There are several new experiments/analyses performed as part of this response. Knowing that these reviews and responses will be archived publicly, in some places we have opted to leave this new work in this document rather than move it to the main manuscript. However, in all cases we are happy to include them if the reviewers or the editor thinks they are useful to the story. Below we respond to each comment inline:

Reviewer #1 (Remarks to the Author):

The manuscript “Generating high-quality libraries for DIA-MS with empirically-corrected peptide predictions.” by Searle et al. is a very innovative study in which the authors take advantage of sophisticated predicted peptide spectral libraries that can be even further refined by empirical data from DIA chromatogram libraries. This work combines various new modern approaches in working with DIA data so it certainly is a very valuable manuscript. In fact, the discussions about library approaches and DIA data are of great need in the field of DIA proteomics, so this study is very timely. Major points that the authors make are that this approach described here, reduces or eliminates the need for offline fractionation, and as the authors mention from their experiences is more efficient than other approaches, or approaches that search DIA data directly without library (e.g. Pecan). The authors also mention these new high-quality data bases require potentially less stringent FDR or have less false positives.

Comments:

- A minor point: When referring to MS acquisitions I would recommend to change run to acquisition which is a somewhat more formal, more appropriate way to describe this for a written manuscript.

We have changed “run” to “acquisition” throughout the text.

- In the supplement there is a nice description in tutorial style how to use Prosit and how to generate the libraries and process further for the empirical corrections. However it would be nice if in the main manuscript it could be discussed somewhat more in detail how exactly the empirical correction is performed conceptionally. This is described very sparsely in the main manuscript while it is the heart to the study. For example, on page 3 the authors mention some of the workflow. The authors mention: “We also replace the retention times and fragmentation patterns in the predicted library with sample- and instrument-specific values found in the 6 runs.” Is this a full replacement or an adaptation. Also, if the RT is replaced does that mean that in the lab that will be using the predicted libraries, the user needs to acquire their GPF acquisitions (or chromatographic libraries for empirical correction) right before their real sample DIA acquisitions? Likely not; but what should be paid attention to. Could the authors elaborate in more detail the chromatographic retention time (RT) aspect of the empirical correction. The MSMS fragmentation adjustments seem more intuitive and clear; however as the RT aspects of the correction seem to be so crucial – does it matter at what time the user lab acquires their chromatographic libraries – or will this also work if they acquired those much earlier in time, and then align for that later---what are aspects the user has to pay attention to. Some form of deeper than provided visualization of the workflow would be beneficial.

We agree with the reviewer and we have thoroughly extended the section titled “Generating empirically-corrected libraries from peptide predictions” from 1 paragraph to 4 paragraphs to more thoroughly describe the overall approach. We have also changed the original Figure 1a to a more detailed independent Figure 1 (see below). In this new text we specifically answer the reviewers questions:

- **How does RT and fragmentation replacement work?** We added: “First, assuming that virtually every consistently quantifiable peptide in an experiment is detectable from the pool using GPF-DIA, we filter the predicted library to remove peptides that cannot be found in the pool. In addition, we select the highest scoring (and therefore most easy to detect) charge state for each peptide and remove other lower scoring charge states from the library. Then, for each identified peptide, we calculate the aggregate peak shape across all of the identified fragment ions and then extract fragment peak area intensities for all possible B- or Y-type ions that correlate with this shape. Since the GPF-DIA injections are performed using the same instrumentation setup as the single-injection DIA injections, we use these intensities as the fragmentation patterns in the empirically-corrected library. Similarly, we use the time point of the apex intensity of the aggregate peak shape as the retention time in the new library.”
- **Do the GPF-DIA runs need to be acquired right before the real sample acquisitions?** We added: “We find that while peptide ordering on the same HPLC platform with the same column and method is typically very high, we still benefit from retention time alignment to account for fluctuations in run-to-run chromatography stability. In addition, we run the 6 GPF-DIA runs of the pool near the middle of an experiment after at least one full set of biological replicates to limit variability caused by column (re)conditioning to a new proteome composition. We recommend reacquiring the 6 GPF-DIA runs if the column or gradient change while an experiment is being conducted.”

- In the methods section on page 18, there is a section called “Empirically-corrected library generation” which is very short; again while this is really what the entire paper is based on – it would be good to elaborate a bit more.

Similar to above, we have extended this section and included more details about how the peak shapes get estimated and how fragment ion peaks are filtered. We also include more discussion about how the manual validation for missense variants was performed. We thank the reviewer and feel that both the new in-text and methods sections improve the manuscript.

- The manuscript then describes several examples where this approach was applied, however, I think for the method of this approach maybe a workflow that is more detailed than the current small Figure 1a would be very helpful. That can bring out key points such as use of the empirical database primarily to detect peak groups; while then the correction uses ‘in house’ (or also other?) chromatographic libraries so that in the end for the corrected library fragmentation and RT are derived from the chromatographic library solely?

As the reviewer requested, we have significantly expanded the original Figure 1a to a new Figure 1. We feel that the complexity of this new figure warrants moving it out as an independent figure. We appreciate this suggestion and feel it improves the manuscript. The new figure is:

Figure 1: Workflow for generating empirically-corrected libraries. Fragmentation patterns and indexed retention times (iRTs) are generated with Prosit for all possible tryptic peptides in a FASTA database and these predictions are compiled into a predicted spectrum library. In this example, peptides from CDPK2 are shown with start/stop indices within the protein indicated in parentheses (red). We use EncyclopeDIA to search GPF-DIA acquisitions of a sample pool with that library, and peptide detection results are compiled into a experiment-specific, empirically-corrected library. This new library contains fragmentation patterns and retention times extracted from the GPF-DIA data for only the detected peptides (blue). Since GPF-DIA and single-injection DIA have the same instrumentation and on-column matrix, retention times and fragmentation patterns in the empirically-corrected library are more accurate than the original predictions.

- In Figure 1d the authors state that there is a big discrepancy between peptide detection using either a predicted library or an empirically corrected library particularly at the lower end of the NCE (the authors state that the NCE is not optimized at the lower end), but could this be explained a bit more possibly?

We have clarified this in the manuscript with the following text (note, Figure 1d is now 2c):

“Fewer peptides will be detectable in both single-injection DIA and GPF-DIA data at incorrect NCE settings. However, since there is less interference in GPF-DIA, these detection rates do not drop as quickly. After empirical correction, the library will contain fragmentation patterns observed in the GPF-DIA data rather than the original library tuning parameters (Supp. Fig. 1), and any peptide that can be detected in the GPF-DIA data will be easier to detect in single-injection DIA. In this way, the GPF-DIA functions as a calibration step that fixes the NCE setting in the library, making searches of single-injection DIA less sensitive to prediction accuracy after empirical correction (Fig. 2c).”

- At the end of the manuscript maybe in the discussion it would be helpful if the authors could indicate what would be the strategies who to implement this workflow into the users own lab. What are the steps the users need to do in order to make this workflow work in their own lab so it becomes a useful tool for users. Would the user need to acquire and generate their own chromatographic library with GPF – would other sample-specific chromatographic libraries generated by the user from their own samples work potentially also (if they are not GPF, maybe a nice variable window – built library would also work) ? Can the users use other people’s chromatographic libraries ? What steps can the authors recommend to users how to implement this for their own work – this could be discussed or presented in a schematic overview. Even if not all answers are known, it would be a good thing to discuss.

Rather than put this in the discussion, we thought it would be better to demonstrate library re-use in the main text. We added a new Supplementary Figure 4 to help address this question:

The discussion in the main text is as follows: “We were interested to determine if empirically-corrected libraries could be reused for different experiments. To test this, we reanalyzed yeast datasets¹³ from a Thermo QE-HF MS using the empirically-corrected library generated in this study on a Thermo Fusion Lumos at a different location. We found that while the empirically-corrected library could be reused to analyze data on a different instrument if GPF-DIA injections were not collected, best results were produced if the additional GPF-DIA injections were collected on the same instrument (Supp. Fig. 5). In this case, collecting additional GPF-DIA injections and building an empirically-corrected library for each experiment improved peptide detection rates by 30%.”

- As part of this manuscript data is uploaded or being re-used and this is reported; Prosit and EncyclopeDIA are also mentioned with their weblink – but are the empirically corrected libraries also provided on some repository – this would be good to have.

We have uploaded these libraries to the original Massive MSV000084000 dataset as “Spectral Libraries”.

- In Figure 3 there are some Venn diagrams and overlap of detection numbers using various different approaches; do the authors think this may have to do with use of different instrument platforms (Lumos, QE ? – it gets difficult to keep track what was acquired on what platforms) in some of those used data sets (in the Venn diagram).

In general, we felt that the overlap between datasets was surprisingly high. The DIA and DDA methods were conducted using the same Thermo QE-HF at the same time, so they can be directly compared. The Lasonder *et al.* study (Lasonder *et al.* 2016) was conducted using a Thermo 7 Tesla LTQ-FTICR. While this is an older instrument, the Lasonder *et al.* study remains the most comprehensive study on the sexual *P. falciparum* proteome to date, in part due to its use of GeLC-MS to fractionate proteins before mass spectrometry.

While the number of proteins detected in the Lasonder *et al.* study reflect other experiments in the literature, the use of older instrumentation is surely an effect. To illustrate this, for another unpublished experiment we analyzed a different *P. falciparum* life stage that had lower contamination from host material with DDA (14% of PSMs on average, versus approximately 30% in our samples). In this case, peptides were pre-fractionated into eight fractions by high-pH reversed phase, each of which was analyzed in technical duplicate with a two-hour gradient using a QE-HF. This approach took 40 hours of data acquisition (16 injections) and we identified a total of 3165 *P. falciparum* proteins from this sample, which is still smaller than the library we present here.

Reviewer #2 (Remarks to the Author):

In their manuscript, Searle et al. describe a workflow to generate spectral libraries for data-independent acquisition mass spectrometry-based proteomics (DIA-MS). DIA-MS has been popularized over the past years and to reach optimal performance, it typically requires empirically acquired spectral libraries, frequently spanning dozen to hundreds of additional data-dependent acquisition (DDA) measurements. Previously, strategies have been proposed to alleviate this burden by generating public repositories and this has worked reasonably well for selected model systems. A bigger issue are applications of DIA-MS to less characterized samples and organisms, where it is typically tedious to acquire such comprehensive spectral libraries.

The authors now combine their previously published spectral library prediction methods with their previously published gas-phase fractionation (GPF) method for library refinement. They demonstrate convincingly that this is a powerful strategy to generate comprehensive spectral libraries that would otherwise be very challenging to acquire. The examples of the HLA variant detection and Plasmodium falciparum illustrate clearly the benefits of their workflow for such applications. I believe that their work can have a considerable impact on how spectral libraries are generated for DIA and that it might hopefully replace DDA-based methods in the future. I thus recommend the paper for publication, however, I have a few minor comments and suggestions.

1. Comparison to deep fractionated libraries

Currently, the highest number of detected peptides and inferred proteins with single injection DIA were reported by using heavily fractionated, repository libraries [<https://www.mcponline.org/content/16/12/2296.long>]. The authors show the performance of the direct application of a 39-injection DDA HeLa library, which I believe is reasonably comprehensive. In the raw comparison, they also show that this library performs better than the unrefined predicted library. But how does the DDA HeLa library perform when it is refined by the same approach? Does the DDA library contain the majority of peptides/proteins contained within the refined predicted library or do they enable detection of different peptides?

We have compared the DIA-only method in our manuscript with the DDA HeLa library refined by the same approach (the chromatogram library refinement). Both methods produce very similar results when using DDA libraries to seed the refinement approach for both peptides (a) and proteins (b):

We have included the peptide comparison for both yeast and HeLa, as well as a comparison to the original chromatogram library method (using Walnut/Pecan) in an updated Figure 3a.

There are some differences between these libraries: the DDA library contains +2H to +6H peptides, as well as oxidized and n-terminal acetylated peptides with up to 2 missed cleavages, while the Prosit library contains only unmodified +2H and +3H peptides with up to 1 missed cleavage. Interestingly, when ignoring PTMs, the Prosit-derived chromatogram library is slightly larger than the DDA-derived chromatogram library. Here is a Venn diagram showing the overlap of unique peptide sequences in the different libraries:

This suggests that in our dataset, being able to use the optimal charge state (e.g. +4H, +5H) for the same peptide sequence may be more important than peptide selection from additional

missed cleavages (different peptide sequences), however, due to the experiment-specific nature of these parameters, this may not be transferable knowledge to experiments in other labs.

2. Supplementary Figure 3

In their original paper, describing the Prosit method to predict spectral libraries, the authors benchmark the predictions extensively. However, after reading the manuscript here, I was surprised that the authors did not provide an assessment of the decomposed effects of their library refinement workflow in the main text. Supplementary Figure 3 contains all information and I believe it would be useful to convince the readers on the performance of each step if it could be displayed as part of Figure 2.

Based on the reviewers suggestion, we have moved Supplementary Figure 3a to Figure 2d so that this point does not get lost in the supplementary materials.

Reviewer #3 (Remarks to the Author):

*The authors propose an approach to generate spectral libraries for analysis of DIA proteomics experiments which proceeds in three main steps: 1) generate a large spectral library of predicted MS/MS spectra using Prosit and a fasta file with protein sequences for the species of interest, 2) acquire new "filtration DIA" data for the sample type of interest and use it to filter the predicted spectral library to contain only matches to the "filtration DIA" runs and 3) use the filtered spectral library to detect peptides in new single-injection DIA runs of the *exact same* sample type. Under these circumstances, the manuscript reports gains in the sensitivity of detection of peptides, proteins and missense sequence variants, but unfortunately it was not possible to thoroughly evaluate these due to the lack of details and supporting information (as described below).*

The first major issue with the manuscript is that (1a) the comparisons between DDA and DIA results are nearly impossible to evaluate and (1b) whatever comparisons appeared possible from the deposited data indicate severe biases against DDA that could suffice to explain the gains reported in the manuscript. Regarding (1a), there is no indication of what deposited raw files were used to derive which set of conclusions in the manuscript so there is no way to properly evaluate whether the results are due to the software analysis or instead due to issues with the acquisition protocols. For example, the deposited data does not seem to include any data for the plasmodium dilution experiments and it's completely unclear which MS runs were used to construct the DDA libraries used as point of reference, or which runs were used to derive the sets of search results reported for the DIA/DDA comparisons. See "other comments" at the end of the review for a suggestion of how this information could be properly specified in a revised version of the manuscript. Regarding (1b), even a cursory analysis of the yeast DDA data in your MSV000084000 dataset reported in the paper reveals that the DDA runs have an average of 5 MS2 scans per MS1 scan (much worse than expected for the top-20 configuration described in online methods, average 48k MS2 spectra per file), whereas the single-injection yeast DIA files in the same dataset have an average of 152k MS2 spectra per DIA file, thereby making it much less surprising that DIA could generate more identifications (as it may be simply due to the acquisition settings generating many more MS2 spectra).

First, some raw files were only made available in an updated version of the MSV000084000 dataset, and MassIVE separates these into different FTP folders. We have rechecked the dataset and made sure that all of the raw files used in this study are publicly available. Unfortunately, we cannot change the MassIVE directory structure to make it more obvious where these files are. However, we have included the MassIVE FTP directory structure for each raw file (as well as what experiment these files are used for) in the new Supplementary Data 3 (see below).

The goal of this manuscript is not to make an argument of DIA versus DDA. Our point is that all libraries benefit from being calibrated for DIA. We have previously demonstrated that DDA libraries work well for DIA after calibration with the chromatogram library method (Searle et al. 2018). Here we show that we can generate DIA-only libraries using calibrated peptide predictions that work as well as calibrated DDA libraries. Calibration with GPF-DIA is important because:

1. DDA requires offline fractionation to produce large peptide libraries. Offline fractionation changes the matrix a peptide sees on column. This causes errors in retention time measurements. Calibrating with GPF-DIA solves this because it is a fractionation method that does not affect the matrix. Theoretically GPF-DDA would solve this, but it has been shown several times that this approach does not perform well in practice (this is best demonstrated by (Ting et al. 2017)).
2. DDA fragmentation is tuned on a peptide-by-peptide basis, which is not done in single-injection DIA. On some instruments it is possible to develop DDA methods that remove this “tuning”, but on Thermo instruments this cannot be done without access to developers tools (and thus not available to most researchers). Calibrating with GPF-DIA solves this because the fragmentation energy is consistent between the methods.

The reviewer brings up an important point about judging the quality of our yeast DDA library. Since yeast is a fairly simple organism, our goal with DDA library construction was to collect as high-quality MS2s as possible to improve our confidence in the library. Consequently, we configured MS2 acquisition to use 30,000 resolution scans with max ion injection time (IIT) of 50 msec, mirroring our DIA library building injections, which were set to 30,000 resolution using a max IIT of 60 msec. In this configuration, overall scan speed can be as slow as 10 hz, frequently forcing our DDA method to collect an MS1 before 20 MS2s could be collected. Considering the number of MS2s per file, the DDA library-generation method collected a median of 48.6k MS2s, while the DIA library-generation method collected a median of 75.8k MS2s. Considering that the DIA method is collecting MS2s regardless of signal (i.e. during the loading and wash steps), this ratio is not surprising.

Here are box plots showing the number of MS2s per MS1 in the Lumos fractions (30,000 resolution):

In addition to the offline high-pH fractionated reverse-phase DDA yeast library collected on a Thermo Fusion Lumos used in this study, we also collected a 6 fraction offline SCX reverse-phase DDA yeast library on a Thermo QE-HF for another study. Here we configured the instrument to use a more standard top-12 DDA method with 15,000 resolution MS2s. As the reviewer noted, the difference between the number of MS2s between the QE-HF and Lumos methods is generally higher. Here are box plots showing the number of MS2s per MS1 in the QE-HF fractions (15,000 resolution):

Indeed, the QE-HF methods hit the top-12 limit more frequently than the Lumos methods hit the top-20 limit, although most of the Lumos injections hit a median of >top-12 at some point during the injection. Unsurprisingly, while we find that these libraries produce about the same number of peptides, these experiments draw from different peptide pools:

Searching the combined DDA library did produce an uptick in the number of detected peptides, mirroring the result from the HeLa experiment where the library was also produced from a mixture of high-pH reverse-phase and SCX fractions.

As such, we conclude that the yeast DDA library used in the paper is a representative DDA library for this sample type. Other fractionation/acquisition methods do not dramatically change the number of library peptides, and we argue that regardless, most labs do not have the capacity to use multiple fractionation approaches. However, combining fractionation methods does increase the library size, resulting in an increase in the number of detected peptides from single-injection runs. We have included this result as a new Supplementary Figure 5, and an updated Figure 3a.

Also related to problem (1b) above, the second major issue with the manuscript (2) is that it does not compare to the state of the art DDA acquisition mode – the BoxCar method published by Meier et al in 2018 [PMID:29735998]. The BoxCar approach was clearly shown to substantially increase DDA sensitivity by reducing the dynamic range limitations of MS1 scans, which are critical for the selection of precursors for MS2 acquisition – a problem that seems to significantly constrain the utility of DDA runs used for this manuscript (as discussed for 1b above). The lack of this comparison is even more glaring since the approach proposed in this manuscript uses similar concepts of gas phase separation and staggered window patterns, thereby severely biasing the comparisons of any DDA and DIA results even before consideration of any software or data analysis steps (which are the main point of the submitted manuscript). While the authors could argue that the BoxCar approach is a recent approach that is not yet widely used, that does not suffice to exclude a comparison with this state of the art DDA approach, as otherwise this manuscript would be unfairly comparing state of the art DIA acquisition with what is essentially a decades-old DDA acquisition mode that is already known to be inferior to the DDA state of the art.

BoxCar is an accurate mass and time (AMT) tag approach, similar to those developed in the early 2000s (reviewed in (Pasa-Tolić et al. 2004)). Both the AMT tag approach and BoxCar only collect MS1 scans. As such, BoxCar is not a “DDA mode” because it does not collect data dependent MS/MS. MaxQuant uses spectral libraries to analyze BoxCar data, where “peptide identifications would be transferred from the library using the ‘match between runs’ feature of MaxQuant, whereas the quantitative information was provided by BoxCar single runs.” (page 444 of (Meier et al. 2018)). There is a strong synergy between BoxCar and our approach that we now include in the Discussion section:

“In addition to DIA applications, this method is applicable for building accurate mass and time tag(Nepomuceno et al. 2003; Pasa-Tolić et al. 2004) libraries for MS1-only data acquisition strategies, such as BoxCar(Meier et al. 2018). This approach forgoes collecting MS/MS and relies on highly accurate mass and retention time indices to identify peptides using “match between runs”. Error rates for “match between runs” peptide detection without MS/MS spectra are both often higher than 1% FDR and hard to estimate without controlled experiments(Lim, Paulo, and Gygi 2019). Errors caused by this approach are likely exacerbated when the on-column matrix changes, such as between fractionated and unfractionated proteomes. Since our approach builds libraries using the same on-column matrix, retention time tags built with GPF-DIA will likely remove this source of variability.”

BoxCar data acquisition can only be performed without service contract voiding firmware modifications on some versions of the Thermo Fusion Lumos firmware and the new Thermo Eclipse/Exploris instruments. As far as we understand, no other instruments or vendors are supported. Unfortunately, we do not have access to an instrument that can acquire data using this method (without violating our service contracts), and we cannot test this synergy directly. For the peptide detection experiments (yeast and HeLa) we do not use match-between-runs for any method (DIA or DDA) to ensure that it is a fair measure of performance in situations without replicates. As such, a comparison to BoxCar from a detection perspective would not fit because it requires match-between-runs to detect any peptides.

The third major issue with the proposed approach is that (3a) it still requires additional MS runs to generate spectral libraries for peptide detection and (3b) it generates spectral libraries that could be extremely sample-specific and thus nearly useless for different sample types, even if from the same species (e.g., different tissues or cell lines). The first part of concern (3a) is the assertion in the manuscript that DDA runs are less useful than DIA runs for pruning predicted spectral libraries. Also related to major issue (2) above, this cannot be properly established without comparing with BoxCar DDA runs to properly assess whether those runs would retain larger subsets of the predicted spectral libraries. In fact, the comparison to DDA should also consider the possibility of retaining predicted spectral library entries if the DDA run (BoxCar or not) contains a LC/MS feature for the precursor m/z within the retention time tolerance, regardless of whether the DDA run generates an MS2 spectrum for the LC/MS feature or not. It would then be important to report how much larger the resulting libraries would be and to assess the impact in loss of sensitivity (if any). The second part of (3a) is that the obvious alternative to requiring additional MS runs (DDA or DIA) would be to use existing spectral libraries as a way to filter precursors in the ProSight predicted spectral library. As such, the authors could filter the yeast predicted spectral library to include only precursors contained in the NIST yeast spectral libraries (>65k distinct precursors over 3 libraries, see

<https://chemdata.nist.gov/dokuwiki/doku.php?id=peptidew:cdownload>) or precursors in the PeptideAtlas yeast spectral library (86k distinct precursors, see <http://www.peptideatlas.org/builds/>). Similarly, the predicted spectral library used for the HeLa searches could be filtered using the entries in the PeptideAtlas human spectral library (1,807,958 distinct precursors) or in the MassIVE-KB human spectral library (2,154,269 distinct precursors). In fact, since MassIVE-KB was generated from Thermo/QExactive data and should thus be compatible with the DIA spectra generated for the results comparisons in this study, the manuscript should also report the results of using MassIVE-KB for peptide detection in the DIA runs when adding predicted retention times to MassIVE-KB precursors. In particular, note that this experiment is conceptually equivalent to replacing predicted spectra with MassIVE-KB spectra while retaining the predicted retention times in the Prosit library – this is also important since the authors report that predicted spectra were not optimal and the results improved when replacing them with real spectra (page 4, lines 8-9). As such, it is important to determine whether using real MS2 spectra from other libraries would also perform better than the predicted MS2 spectra.

Repository-sized libraries are typically considered weaker libraries for DIA analysis due to increased variability when compiling spectra from multiple sources as well as a more stringent FDR correction (both are demonstrated in (Bruderer et al. 2017)). Indeed, Bruder *et al.* advocates for using the most experiment-specific library available to achieve the best quality results. Our goal with this work is to make generating experiment-specific libraries easier, not to generate pan-organismal libraries for global use. We agree that our approach to generating experiment-specific DIA libraries costs an additional 6 MS injections, and we believe that many DIA researchers will be willing to pay this cost over the cost of generating experiment-specific DDA libraries for non-human organisms, or needing to account for library staleness as instruments change. We have added this paragraph to the discussion to clarify our views:

“In conclusion, empirical correction of predicted spectrum libraries enables rapid experiment-specific library generation for non-canonical proteomes or non-model organisms without offline fractionation. DDA-based spectral libraries can become stale over time as columns are changed or NCE tuning drifts within an instrument. While the method we propose to create empirically-corrected libraries requires an extra 6 GPF-DIA injections for each new experiment, the procedure has the advantage of ensuring that the library is always "up-to-date", and even accounts for variation across different instrument platforms.”

The reviewer correctly notes that the repository-sized libraries (MassIVE-KB, PeptideAtlas, NIST, etc) are not designed for most DIA analysis because they do not include relative retention times. While we do not have an easy mechanism to combine the retention times from Prosit with the spectra from MassIVE-KB, we can use it with predicted retention times from another algorithm, SSRCalc3. As expected from the Bruder *et al* insights, this approach performs better than Prosit predictions but worse than the experiment-specific DDA library. The Pan-Human library (Rosenberger et al 2014) is a spectral library generated specifically for DIA analysis and contains real retention times on ToF instruments. While not repository-sized, this library contains >200k spectra collected from >300 injections of different human cell lines (including HeLa) and tissues generated by one lab. We find that of the global library methods, this performs the best with our data:

We have not included this followup analysis in the manuscript, but we are happy to include them if the reviewer or the editor thinks they are useful to the story.

Regarding issue (3b), the expected utility of the proposed method is directly proportional to its generalization to other sample types so it is important to assess how well the filtered spectral libraries would work for different sample types such as different tissues or different cell lines. A simple way to test this would be to re-search the QExactive data from K562 cell lines used for the original DIA-Umpire paper (file 140120_Lysate_90min_DIA_53.raw), which is readily available in the public dataset PXD001587. This file (and hopefully other DIA runs from other human samples) should be used to show how the performance of the DIA-filtered predicted spectral library compares with that of the spectral libraries filtered to contain precursors from PeptideAtlas or MassIVE-KB (including direct searches with predicted retention times). If it turns out that the public-library approach performs comparably in terms of sensitivity on HeLa runs but is much more generalizable to other types of samples, then the manuscript will need to clearly quantify and justify the expected gains from requiring additional instrument time to acquire GPF DIA data solely for filtering predicted spectral libraries.

These libraries are meant to be “experiment-specific” (not “species-specific”), so comparisons across different cell lines and tissue types will not be fruitful. We have added additional clarification about this in the abstract and discussion. With regards to public libraries, as shown above, we find that the empirically-corrected library method produced 20% more detections compared to searching the pan-human library (Rosenberger et al. 2014) in our HeLa datasets. The reviewer brings up an excellent point (also echoed by reviewer 1) about whether empirically-corrected libraries are transferable between instrument platforms, and what penalty the user takes if they reuse a library on a different instrument or in a different lab. To test this, we reanalyzed a previously published yeast dataset collected on a Thermo QE-HF, using the library we generated using a Thermo Fusion Lumos. The yeast samples in both experiments were cell-culture replicates produced by Dr. Lindsay Pino (University of Washington), allowing us to study the effect of instrumentation relatively separate from sample generation. We included this analysis as a new Supplementary Figure 4. Briefly, searching the reused empirically-corrected library from a different instrument platform produced better results than either searching the predicted yeast library directly, or than with Pecan (searching just a FASTA). However, collecting additional GPF fractions on the same instrument platform dramatically improved the detection rates by 30%. This suggests that while empirically-corrected library reuse is possible (and better than using a predicted library as is, we highly recommend collecting GPF-DIA fractions along with an experiment. As long as an experiment contains more

than a handful of samples, we find that these injections In addition to new manuscript text, the new supplementary figure is shown below:

Supplementary Figure 4: Comparison of single-injection DIA peptide detection using different library approaches. Triplicate yeast DIA injections from Searle *et al* 2018 collected with a Thermo QE-HF searched as if no GPF injections were collected (purple), or using GPF injections to build libraries (orange). If the GPF injections were not used, three analysis methods are possible: either searching the single-injection DIA data with Pecan directly, with a predicted library, or using the empirically-corrected library generated from this study on a Thermo Fusion Lumos in a different laboratory. While the reused empirically-corrected library performs better than the other search strategies, the best results are generated when additional GPF injections can be collected on the same instrument, either with the standard chromatogram library method using Pecan, or the empirically-corrected method described here.

The fourth major issue is that there are many missing details about the detection of HeLa variants and some of the reported results appear to be somewhat nonsensical. First, it is important that post-translational modifications (real or artefactual) be considered as possible alternative explanations for the reported variants. MS2 spectra should be provided in supplementary materials (both reference and variant predicted spectra and real spectra if present in DDA data) and it should be discussed how the distinction from possible modifications is made using only the single-injection DIA data (e.g., how-many/which fragments were used, what mass accuracy was required, show spectra/chromatograms, etc). Second, both reference and variant peptide sequences should be provided in supplementary materials. The submitted variants supplementary table has a number of issues: i) some variants are reported to contain mutations to amino acids that do not occur in the sequence (e.g., “A119T” for peptide #13, but the peptide does not contain Threonine); ii) the alternative interpretation that “Peptide Sequence” would mean “reference sequence” also does not hold since peptide #2 is listed with a mutation “I253T” but contains no Isoleucine; iii) the notation “X-number-Z” for amino acid variants is expected to represent a variation of amino acid X to amino acid Z at coordinate “number” in the protein sequence but the numbers reported in the spreadsheet make no sense in this context. For example, peptide #14 is reported to contain two amino acid substitutions, one at coordinate 934 (C934G) and another at coordinate 1354 (R1354H), which could never

occur on the same peptide if the substitutions were over 400 amino acids apart. Since the inconsistency remains even if the coordinates are in base pairs on the original transcript sequence (the substitutions would still be >130 amino acids apart), it is important to correct these to more meaningful amino acid coordinates (preferably on the corresponding UniProt protein sequences). Third, it is surprising for the authors to claim that a single-injection DIA run could detect more variants than 1,206 DDA runs of the same cell line. Since there is a very high chance that this subset of new/surprising variants might be substantially enriched for false discoveries, it is necessary to also list in the supplementary materials table whether each peptide was previously detected or not.

Unlike missense variants that are determined *de novo* from just mass spectrometry data, all of the missense variant peptides that we observe are expected from the COSMIC sequence data. In fact, in some cases the canonical forms are not expected to be present because the genome is homozygous for those SNPs and observing the canonical forms would require justification. Even still, we have generated a new Supplementary Figure 6, which shows chromatograms for all of the 37 missense variant containing peptides we report. For brevity, here are just the first 15 chromatograms shown in the new supplementary figure:

Our table mistakenly annotated all variants in a protein together, while typically only one variant was found per peptide. For example, peptide 13 (MSQEPEINKDCDR to MSQEPEINKDCDK) was incorrectly annotated as “p.M43V;A119T;R413K” instead of just “R413K”. Similarly, peptide 14 (LVERGAPQSLLLSESGK to LVEHGAPQSLLLSESGK) only indicates “R1354H”, not “p.C934G;R1354H”. We have fixed this misannotation in the supplementary table. There is only one example of detecting two variants in one peptide (peptide 34, YFFTSVSRPGR to YFYTSMSRPGR). Since we counted total variant peptides (not total variant sites) this does not change any of the figures. We thank the reviewer for bringing this to our attention.

- Need to provide supplementary materials with full list of files used for each experiment described in the manuscript. In particular, it is important to describe which files were used for construction of spectral libraries and which files were used to derive each set of DDA or DIA results presented in the manuscript. One simple way to achieve this is to refer to each set of results as experiment X (where X could be a number) and then provide a spreadsheet in

supplementary materials linking each raw file to an experiment if the raw file was used to construct the corresponding library (one kind of experiment) or to derive the corresponding DDA or DIA search results (another type of experiment).

We thank the reviewer for pointing out this missing table. We have added a new table as Supplementary Data 3 linking each raw file on MassIVE (accession MSV000084000) with an experiment (yeast or plasmodium) and a data acquisition method. The reused HeLa data was already described in Supplementary Data 4 of (Searle et al. 2018).

- Figure 1d shows that Prosit prediction at NCE 36 was the best match to the experimental NCE 27 used for the MS runs – was that the case for all libraries? If yes then should it be concluded that the Prosit NCE parameters need to be recalibrated?

Prosit is trained to estimate NCE for Thermo Fusion instruments. Although the Thermo Fusion Lumos instrument used in this experiment was configured to NCE=33, here the NCE=36 predicted library performs slightly better. This subtle difference reflects minor calibration variances between different instrument copies, and we have observed small variance in NCE calibration even amongst different copies of the same instrument model in the same lab (Zolg et al. 2017). We now point this out specifically in the manuscript. Interestingly, even on the same instrument, NCE varies over time. We have seen differences in NCE of up to 4 on the same mass spectrometer as it ages.

We configured the Thermo QE-HF to NCE=27, which again reflects an internal calibration issue in different types of Thermo instruments (i.e. those made in Bremen versus those made in San Jose). We have found that in order to achieve the same fragmentation quality, there is typically an approximate 6 NCE difference between Fusion-class instruments and QE-class instruments for most peptides (described in Zolg et al). Here we used Prosit libraries calibrated for NCE=33 (trained on a Fusion) to match the QE-HF configured at NCE=27.

- The retention time predictions also seemed to be inaccurate (80% of peptides within 5.4 minutes) and the variation between the DDA and DIA retention times also seems very high (80% of peptides within 4.6 minutes), whereas the matches of retention times between GPF runs and single-injection DIA runs was much better (80% of peptides within 35 seconds). What is the rationale for the dramatically better match of retention times between GPF/single-injection DIA runs than between DDA/DIA runs? If all MS runs were generated under the same chromatographic conditions then there should not be such dramatic variation in the retention times between DDA and DIA runs. Of particular importance, how much worse would the search results be if the GPF-DIA-filtered predicted spectral libraries had to deal with the same retention time errors as the libraries derived from DDA data? How can it be quantified how much of the improved performance comes from correcting retention time disparities that should not exist in the first place (i.e., in DDA vs DIA runs; the disparities are understandable in relation to predicted retention times)?

There are two important questions here, 1) how much does retention time accuracy help, and 2) why is retention time accuracy in the DDA library poor? First, the breakdown of how much retention time alone affects detection rate is illustrated in the new Figure 2d (originally Supplementary Figure 3a). The fully empirically-corrected library performs 13% better than the

partial predicted library using empirical fragmentation and peptide selection, but predicted retention times.

Second, retention time is affected by chromatographic conditions, but also by matrix effects. Single-injection DIA and DDA both measure peptides using the full matrix. GPF-DIA uses the quadrupole for fractionation, and consequently also measures peptides using the full matrix. Offline fractionation, such as SCX or high-pH reverse-phase change the matrix by fractionating the peptide mixture into multiple samples. Consequently, each peptide sees a different matrix as it elutes, as compared to the single-injection runs. This is the reason why the fractionated DDA injections show retention time variability, while the GPF-DIA injections do not. We have made this more clear in the discussion of Figure 2d and Supplementary Figure 2.

- It was not possible to access the spectral libraries at ProteomicsDB (<https://www.proteomicsdb.org/prosit/libraries>) since the link is not accessible via free anonymous proxy services. All missing data and all full and filtered spectral libraries should be uploaded to a new dataset at one of the ProteomeXchange repositories.

The ProteomicsDB link is now online and freely accessible. This website is for unfiltered libraries so that other people can use them to make “calibrated” libraries for their instruments. Currently we have a limited number of libraries uploaded there, but the list is continuing to grow. We have also added the calibrated libraries generated for this study to the original MassIVE MSV000084000 dataset.

- The proposed approach is conceptually similar to the approach used by MSPLIT-DIA[PMID:26550773] where there is a first stage of analysis that reduces the size of a large spectral library at a fixed FDR and then there is a second stage that uses the reduced library for peptide detection (e.g., see “MSPLIT-DIA library” results reported in Supp. Fig. 9). The approaches should be compared in the manuscript and it should be discussed why the proposed approach is preferable to that of MSPLIT-DIA.

The approach used by MSPLIT-DIA is to search data from a single DIA injection without retention time filtering, and then reanalyze the same DIA injection afterwards using sample-specific retention time filtering to increase signal-to-noise. While this had been possible earlier using indexed retention times, MSPLIT-DIA was able to perform this filtering using unknown peptides. This approach has already been implemented in EncyclopeDIA and discussed in depth in (Searle et al. 2018). However, MSPLIT-DIA is not designed to correct retention times across multiple raw files, or to build DIA-only libraries. We have included a discussion of this in the introduction: “Proteins show 3-4 orders of magnitude difference in intensity between the best- and worst-responding tryptic peptides,(Searle et al. 2015) and only considering the best-responding peptides in libraries can improve detection rates by lessening the required FDR correction. This approach has been applied by generating independent assay libraries for each DIA injection,(Tsou et al. 2015; Wang et al. 2015) and we extend this approach by building experiment-specific libraries from GPF-DIA injections using the same acquisition parameters, chromatographic conditions, and sample matrix as quantitative single-injection DIA experiments.”

- A more extensive discussion of controls for false discovery rates (FDRs) needs to be included in supplementary materials, especially as it relates to demonstrating that the gains in

identifications are not obtained at the cost of elevated FDR. In particular, while the authors do mention that the filtered libraries are generated using EncyclopeDIA at 1% peptide- and protein-level FDRs, it is not clear that any of this holds when running the downstream analyses of single-injection DIA data. This process needs to be described in detail and needs to include a discussion about why and how FDR controls remain valid for single-injection DIA search results. In particular, if the downstream analyses of single-injection DIA data use peptide-centric FDR estimates then it needs to be shown that these are comparable to the spectrum-centric FDR estimates used for the DDA results (otherwise the difference in results could be largely due to just this issue).

We thank the reviewer for pointing out this needed discussion. In addition to significantly increasing the discussion of FDR in library building in the Methods section “Empirically-corrected library generation”, we have added a new section called “False discovery rate estimation”. Much of the concerns of DIA versus DDA FDR estimation are mitigated because we use standard target/decoy practices and Percolator 3.1 (a tool designed for analyzing DDA datasets) to estimate FDRs. After library generation, FDR estimation for single-injection DIA runs is performed twice: once at the individual run level, and again globally across all quantitative samples. For peptide detection experiments, the “match-between-runs” approach was not used. For the quantitative *P. falciparum* experiments, match-between-runs is applied for peptides not detected in every run, but only if the peptide was detected at a 1% FDR in the global analysis and at a 1% FDR in at least one individual run.

- It appears from the manuscript that the Plasmodium single-injection files were searched using only predicted spectral libraries (filtered or not) from Plasmodium sequences. Since the dilution experiments in human red blood cells are obviously also going to contain human peptides, it is necessary to also report results when human sequences are also used a) for the construction of the filtered library and b) for the searches of single-injection dilution runs. In particular, what would be the impact in the loss of sensitivity for plasmodium peptides? Would any DIA fragments/chromatograms switch from plasmodium peptides to blood peptides, especially if searching the same data only against filtered spectral libraries of only blood peptides? These questions are especially relevant in the context of the dependencies that are ignored for peptide-centric analyses that seem to be central to this manuscript so it is necessary to conduct these experiments and to clearly show that the reported gains are not mostly the result of poor FDR controls for false discoveries.

While spectrum-centric analysis requires that there is peptide competition between all likely candidates for each MS/MS, peptide-centric analysis (Ting et al. 2015) does not require competition as long as peptides are not likely to share sequences. In this case we assumed that there would be a minimal amount of shared peptides between Human and Plasmodium, and we thank the reviewer for having us double check this assumption. We performed a similar experiment on GPF-DIA injections of uninfected RBCs from both the perspective of searching for Plasmodium peptides. We observed zero peptides (and proteins) detected when searching the uninfected RBCs with Plasmodium peptides. We have added the following text to the main manuscript: “We performed the same approach of searching for *P. falciparum* peptides on GPF-DIA injections of uninfected red blood cells (RBCs) and found that the library produced 0 peptide and protein detections.” We have also added these GPF-DIA runs to the original MassIVE MSV000084000 dataset.

References:

- Bruderer, Roland, Oliver M. Bernhardt, Tejas Gandhi, Yue Xuan, Julia Sondermann, Manuela Schmidt, David Gomez-Varela, and Lukas Reiter. 2017. "Optimization of Experimental Parameters in Data-Independent Mass Spectrometry Significantly Increases Depth and Reproducibility of Results." *Molecular & Cellular Proteomics: MCP* 16 (12): 2296–2309.
- Lasonder, Edwin, Sanna R. Rijpma, Ben C. L. van Schaijk, Wieteke A. M. Hoelijmakers, Philip R. Kensche, Mark S. Gresnigt, Annet Italiaander, et al. 2016. "Integrated Transcriptomic and Proteomic Analyses of *P. Falciparum* Gametocytes: Molecular Insight into Sex-Specific Processes and Translational Repression." *Nucleic Acids Research* 44 (13): 6087–6101.
- Lim, Matthew Y., João A. Paulo, and Steven P. Gygi. 2019. "Evaluating False Transfer Rates from the Match-between-Runs Algorithm with a Two-Proteome Model." *Journal of Proteome Research*, October. <https://doi.org/10.1021/acs.jproteome.9b00492>.
- Meier, Florian, Philipp E. Geyer, Sebastian Virreira Winter, Juergen Cox, and Matthias Mann. 2018. "BoxCar Acquisition Method Enables Single-Shot Proteomics at a Depth of 10,000 Proteins in 100 Minutes." *Nature Methods* 15 (6): 440–48.
- Nepomuceno, Angelito I., David C. Muddiman, H. Robert Bergen, James R. Craighead, Michael J. Burke, Patrick E. Caskey, and Jonathan A. Allan. 2003. "Dual Electrospray Ionization Source for Confident Generation of Accurate Mass Tags Using Liquid Chromatography Fourier Transform Ion Cyclotron Resonance Mass Spectrometry." *Analytical Chemistry* 75 (14): 3411–18.
- Pasa-Tolić, Ljiljana, Christophe Masselon, Richard C. Barry, Yufeng Shen, and Richard D. Smith. 2004. "Proteomic Analyses Using an Accurate Mass and Time Tag Strategy." *BioTechniques* 37 (4): 621–24, 626–33, 636 passim.
- Rosenberger, George, Ching Chiek Koh, Tiannan Guo, Hannes L. Röst, Petri Kouvonen, Ben C. Collins, Moritz Heusel, et al. 2014. "A Repository of Assays to Quantify 10,000 Human Proteins by SWATH-MS." *Scientific Data* 1 (September): 140031.
- Searle, Brian C., Jarrett D. Egertson, James G. Bollinger, Andrew B. Stergachis, and Michael J. MacCoss. 2015. "Using Data Independent Acquisition (DIA) to Model High-Responding Peptides for Targeted Proteomics Experiments." *Molecular & Cellular Proteomics: MCP* 14 (9): 2331–40.
- Searle, Brian C., Lindsay K. Pino, Jarrett D. Egertson, Ying S. Ting, Robert T. Lawrence, Brendan X. MacLean, Judit Villén, and Michael J. MacCoss. 2018. "Chromatogram Libraries Improve Peptide Detection and Quantification by Data Independent Acquisition Mass Spectrometry." *Nature Communications* 9 (1): 5128.
- Ting, Ying S., Jarrett D. Egertson, James G. Bollinger, Brian C. Searle, Samuel H. Payne, William Stafford Noble, and Michael J. MacCoss. 2017. "PECAN: Library-Free Peptide Detection for Data-Independent Acquisition Tandem Mass Spectrometry Data." *Nature Methods* 14 (9): 903–8.
- Ting, Ying S., Jarrett D. Egertson, Samuel H. Payne, Sangtae Kim, Brendan MacLean, Lukas Käll, Ruedi Aebersold, Richard D. Smith, William Stafford Noble, and Michael J. MacCoss. 2015. "Peptide-Centric Proteome Analysis: An Alternative Strategy for the Analysis of Tandem Mass Spectrometry Data." *Molecular & Cellular Proteomics: MCP* 14 (9): 2301–7.
- Tsou, Chih-Chiang, Dmitry Avtonomov, Brett Larsen, Monika Tucholska, Hyungwon Choi, Anne-Claude Gingras, and Alexey I. Nesvizhskii. 2015. "DIA-Umpire: Comprehensive Computational Framework for Data-Independent Acquisition Proteomics." *Nature Methods* 12 (January): 258.
- Wang, Jian, Monika Tucholska, James D. R. Knight, Jean-Philippe Lambert, Stephen Tate,

Brett Larsen, Anne-Claude Gingras, and Nuno Bandeira. 2015. "MSPLIT-DIA: Sensitive Peptide Identification for Data-Independent Acquisition." *Nature Methods* 12 (12): 1106–8.

Zolg, Daniel Paul, Mathias Wilhelm, Peng Yu, Tobias Knaute, Johannes Zerweck, Holger Wenschuh, Ulf Reimer, Karsten Schnatbaum, and Bernhard Kuster. 2017. "PROCAL: A Set of 40 Peptide Standards for Retention Time Indexing, Column Performance Monitoring, and Collision Energy Calibration." *Proteomics* 17 (21). <https://doi.org/10.1002/pmic.201700263>.

REVIEWERS' COMMENTS:

Reviewer #1 (Remarks to the Author):

The authors have made a great effort to address the concerns of the reviewers. They added several new figures and comparisons that are very helpful

The use of the word 'run' for acquisition should be considered to be changed to acquisition instead of run throughout the manuscript. This is a formal manuscript that should reflect a more scientific and technical language.

REVIEWERS' COMMENTS:

Reviewer #1 (Remarks to the Author):

The authors have made a great effort to address the concerns of the reviewers. They added several new figures and comparisons that are very helpful

The use of the word 'run' for acquisition should be considered to be changed to acquisition instead of run throughout the manuscript. This is a formal manuscript that should reflect a more scientific and technical language.

We thank the reviewer for these comments. We have changed 'run' to "injection" throughout the manuscript. We have opted for "injection" instead of "acquisition" to avoid redundancy with DIA, e.g. Data Independent Acquisition (DIA) Acquisition.